# CONTINUOUS MULTINOMIAL LOGISTIC REGRESSION FOR NEURAL DECODING

**Anuththara Rupasinghe & Jonathan W. Pillow**
Princeton Neuroscience Institute, Princeton University
`{ar0621,pillow}@princeton.edu`

## ABSTRACT

Multinomial logistic regression (MLR) is a classic model for multi-class classification that has been widely used for neural decoding. However, MLR requires a finite set of discrete output classes, limiting its applicability to settings with continuous-valued outputs (e.g., time, orientation, velocity, or spatial position), which are common in neuroscience settings. To address this limitation, we propose Continuous Multinomial Logistic Regression (CMLR), a generalization of logistic regression to continuous output spaces. CMLR represents a novel exponential-family model for conditional density estimation (CDE), mapping neural population activity to a full probability density over external covariates. It captures the influence of each neuron's activity on the decoded variable through a smooth, interpretable tuning function, regularized by a Gaussian process prior. The resulting nonparametric decoding model flexibly captures asymmetric and multimodal densities, and accommodates both linear and circular variables. To illustrate the performance of CMLR, we applied it to large-scale datasets from mouse and monkey visual cortex, mouse hippocampus, and monkey motor cortex, where it generally outperformed a wide variety of other decoding methods, including deep neural networks (DNNs), XGBoost, and FlexCode. It also outperformed a closely-related correlation-blind decoder, highlighting the importance of correlations for accurate neural decoding. The CMLR model provides a scalable, flexible, and interpretable method for decoding continuous variables from diverse brain regions.

## 1 INTRODUCTION

Neural decoding refers to the problem of estimating behavioral or sensory variables from neural activity, a central challenge in neuroscience (Georgopoulos et al., 1982; Brown et al., 1998; Zhang et al., 1998). Logistic regression is a foundational model for binary classification that has been widely applied to neural decoding problems with two alternatives (Ryali et al., 2010; Glaser et al., 2020). For tasks with multiple alternatives, multinomial logistic regression (MLR) provides a natural extension, defining class probabilities based on the linear projection of neural activity onto weight vectors, one for each output class (Huttunen et al., 2013; Song et al., 2014; Greenidge et al., 2024). However, many neural decoding tasks involve continuous variables such as time, orientation, head direction, position, or velocity (Paninski & Cunningham, 2018). In such settings, standard regression models are frequently inadequate because they yield only point predictions and cannot represent multimodal, circular, or asymmetric output distributions (Hyndman et al., 1996). To obtain full predictive distributions, researchers often adapt MLR-style classifiers by discretizing the continuous output into bins (Stringer et al., 2021; Greenidge et al., 2024). However, this discretization reduces effective resolution, introduces quantization artifacts, and typically necessitates additional regularization to prevent overfitting (Altman & Royston, 2006; Nojavan A. et al., 2017).

To overcome this limitation, we introduce the Continuous Multinomial Logistic Regression (CMLR) model, which generalizes MLR to continuous output spaces. Whereas MLR defines a discrete probability distribution over a finite set of output classes using a log-linear combination of weight vectors, the CMLR model defines a probability density function (pdf) over a *continuous* output variable using a log-linear combination of weight *functions*. These weight functions are conceptually analogous to neural tuning curves, as they characterize how each neuron's activity influences the predicted density over the variable of interest and often resemble the underlying tuning, although they

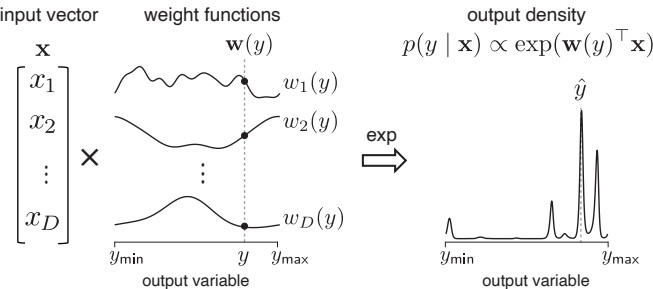

Figure 1: Continuous Multinomial Logistic Regression (CMLR) model schematic. Given an input feature vector $\mathbf{x} \in \mathbb{R}^D$, the model defines a logistic density over $y$ using weight functions $\mathbf{w}(y) = \{w_d(y)\}_{d=1}^D$. The probability density at output value $y$ given $\mathbf{x}$ is proportional to $\exp(\mathbf{w}(y)^\top \mathbf{x})$. For each feature $d$, the function $w_d(y)$ has an independent Gaussian process prior to induce smoothness.

originate from a decoding model rather than an encoding model. CMLR thus provides a framework for conditional density estimation (CDE), a problem setting that involves mapping a feature vector to a probability density (Hyndman et al., 1996). For neural decoding applications, this density can be summarized by its mean or mode to obtain point estimates of the external variable of interest.

To enforce smoothness, we place Gaussian process (GP) priors on the weight functions, an approach closely related to Logistic Gaussian Process density estimation (Tokdar et al., 2004; Riihimäki & Vehtari, 2014). Given a vector of neural activities, the resulting model can represent complex conditional densities over the output variable, including multimodal, asymmetric, and circular forms often observed in neural decoding. To make inference tractable, we develop a Fourier-domain stochastic variational inference algorithm (Hoffman et al., 2013), leveraging GP kernel stationarity for diagonalization and spectral truncation (Hensman et al., 2018). The resulting approach handles high-dimensional inputs, extends naturally to multivariate-output CDE via anisotropic kernels (Rasmussen & Williams, 2006), and handles large-scale neural datasets.

We evaluate CMLR on large-scale datasets from mouse and monkey primary visual cortex (V1) (Stringer et al., 2021; Graf et al., 2011), mouse hippocampus CA1 (Hazon et al., 2022; Jercog & Schnitzer, 2022), and monkey motor cortex (Glaser et al., 2018; 2020). Across datasets, CMLR generally outperforms strong decoding baselines, including Extreme Gradient Boosting (XGBoost) (Chen & Guestrin, 2016), deep neural networks (DNNs) (Orbach, 1962), and the leading non-parametric CDE method FlexCode (Izbicki & Lee, 2017), while producing well-calibrated posterior densities. These advantages are strongest in low-data regimes, where CMLR's GP-based functional priors and additive structure regularize strongly and limit overfitting, and in tasks with structured or multimodal outputs, such as circular orientation decoding, where periodicity and multimodality must be accurately represented. In these settings, access to the full conditional density allows CMLR to capture the underlying structure far more faithfully than point-estimate models such as XGBoost and DNNs, which cannot naturally model circular or multimodal output distributions. Furthermore, extending prior comparisons of correlation-aware and correlation-blind decoders (Nirenberg & Latham, 2003; Graf et al., 2011; Greenidge et al., 2024; Wei et al., 2024), we show that CMLR (correlation-aware), by modeling shared variability, surpasses Naive Bayes (correlation-blind) in predictive accuracy across V1, CA1, and motor cortex. These results establish CMLR as a flexible, interpretable, and scalable framework for high-resolution neural decoding across brain regions.

## 2 CONTINUOUS MULTINOMIAL LOGISTIC REGRESSION (CMLR) MODEL

The CMLR model (schematized in Fig. 1) defines a mapping from an input vector $\mathbf{x} \in \mathbb{R}^D$ (e.g., a vector of spike counts) to a probability density over an output variable $y \in \Omega$, defined over some compact domain $\Omega$. The model parameters consist of a set of $D$ weight functions, one for each element of $\mathbf{x}$, where $w_d(y)$ describes the additive influence of input feature $x_d$ on the log-density over $y$. The resulting conditional density is given by:

$$p(Y = y \mid \mathbf{x}) = \frac{\exp\left(\mathbf{w}(y)^\top \mathbf{x}\right)}{\int_\Omega \exp\left(\mathbf{w}(y')^\top \mathbf{x}\right) dy'}, \tag{1}$$

where $\mathbf{w}(y) = [w_1(y), \ldots, w_D(y)]^\top$ denotes the vector obtained by evaluating all weight functions at $y$, and the denominator provides the normalizing constant, ensuring the density integrates to 1 (Tokdar et al., 2004; Riihimäki & Vehtari, 2014).

Note that this generalizes standard multinomial logistic regression (MLR) with $K$ discrete classes, where the probability of class $k$ is given by:

$$p(Y = k \mid \mathbf{x}) = \frac{\exp(\mathbf{w}_k^\top \mathbf{x})}{\sum_{j=1}^{K} \exp(\mathbf{w}_j^\top \mathbf{x})}, \tag{2}$$

where $\mathbf{w}_k$ denotes the weight vector for class $k$. CMLR can thus be viewed as the continuous limit of MLR as $K \to \infty$, replacing a set of ordered discrete class weights with smooth weight functions defined over a continuous output space.

To enforce smoothness in the decoding weights $\mathbf{w}(y)$, we place an independent Gaussian Process (GP) prior on each weight function:

$$w_d(y) \sim \mathcal{GP}\left(\mathbf{0}, K_d\right), \tag{3}$$

where $K_d$ is the covariance function governing the weight function: $\mathrm{cov}(w_d(y'), w_d(y'')) = K_d(y', y'')$ for any pair of output values $y', y'' \in \Omega$. Here, we employ the standard radial basis function (RBF) covariance (Rasmussen & Williams, 2006): $K_d(y', y'') = \rho_d \exp\left(-(y' - y'')^2 / 2\ell_d^2\right)$, where $\rho_d$ and $\ell_d$ are hyperparameters controlling the marginal variance (i.e., amplitude) and length scale (i.e., smoothness) of the weight function, respectively. For circular output spaces (e.g., orientations with $\Omega = [0, 2\pi)$), we use the periodic version of the RBF kernel defined as (Schölkopf & Smola, 2001): $K_d(y', y'') = \rho_d \sum_{m=-\infty}^{\infty} \exp\left(-(y' - y'' + 2\pi m)^2 / (2\ell_d^2)\right)$, ensuring continuity around the unit circle.

## 3 EFFICIENT LEARNING VIA STOCHASTIC VARIATIONAL INFERENCE (SVI)

Given a dataset of $N$ input-output pairs $\mathcal{D} = \{(\mathbf{x}_n, y_n)\}_{n=1}^{N}$, we aim to jointly fit the weight functions $\mathbf{w}(y) = \{w_d(y)\}_{d=1}^{D}$ and the hyperparameters $\theta = \{\rho_d, \ell_d\}_{d=1}^{D}$ governing the GP prior over each weight function. The joint log-probability of the observations and the decoding weights is given by:

$$\log p_\theta\left(\mathcal{D}, \mathbf{w}(y)\right) = \sum_{n=1}^{N} \left(\mathbf{w}\left(y_n\right)^\top \mathbf{x}_n - Z_n\right) - \frac{1}{2} \sum_{d=1}^{D} \left(\log |K_d| + \mathbf{w}_d^\top K_d^{-1} \mathbf{w}_d\right) \tag{4}$$

where $Z_n = \log\left(\int_\Omega \exp\left(\mathbf{w}(y)^\top \mathbf{x}_n\right) dy\right)$ is the normalizing constant; $\mathbf{w}_d = [w_d(y_1), w_d(y_2), \ldots, w_d(y_N)]^\top$ denotes the weight function of input feature $d$ evaluated at the observed outputs $\{y_n\}_{n=1}^{N}$; and $K_d \in \mathbb{R}^{N \times N}$ is the corresponding GP covariance matrix over these outputs.

This formulation is intractable due to the need to marginalize over weight functions when computing each $Z_n$. To jointly infer the latent functions $\mathbf{w}(y)$ and hyperparameters $\theta$, we thus introduce an efficient inference scheme based on Variational Inference (VI) (Beal, 2003; Jordan et al., 1999; Hoffman et al., 2013; Blei et al., 2017). VI approximates the true intractable posterior distribution $p(\mathbf{w}|\mathbf{x})$ with a tractable distribution $q(\mathbf{w})$, by optimizing the Evidence Lower Bound (ELBO):

$$\mathcal{L}(\theta, \psi) = \mathbb{E}_{q_\psi}\left[\log p_\theta\left(\{\mathbf{x}_n\}_{n=1}^{N} | \mathbf{w}\right)\right] - D_{KL}\left(q_\psi(\mathbf{w}) || p_\theta(\mathbf{w})\right), \tag{5}$$

where $D_{KL}(q||p)$ denotes the Kullback–Leibler divergence between the variational distribution and the prior (Blei et al., 2017). This ELBO serves as the objective that CMLR maximizes during training.

### 3.1 RIEMANN INTEGRAL APPROXIMATION OF THE NORMALIZING CONSTANT

To compute the ELBO, we approximate the intractable normalizing integral in $Z_n$ using Riemann integration. We partition the output range $\Omega = [y_{\mathsf{min}}, y_{\mathsf{max}}]$ into $T$ uniform bins of width $\Delta = (y_{\mathsf{max}} - y_{\mathsf{min}})/T$, and let $\overline{y}_t$ denote the center of the $t^{\mathrm{th}}$ bin. The integral is then approximated as:

$$\int_\Omega \exp\left(\mathbf{w}(y)^\top \mathbf{x}_n\right) dy \approx \Delta \sum_{t=1}^{T} \exp\left(\mathbf{w}\left(\overline{y}_t\right)^\top \mathbf{x}_n\right). \tag{6}$$

## 3.2 Fourier-domain representation of weight functions

To improve computational efficiency, we parameterize the decoding weights in the frequency domain, exploiting the fact that the RBF covariance is diagonalized in a Fourier basis. This allows efficient inference by eliminating matrix inversions and operating in a reduced-dimensional space (Hensman et al., 2018; Gondur et al., 2024; Keeley et al., 2020; Aoi & Pillow, 2017). Let $\mathbf{B} \in \mathbb{R}^{N \times M}$ and $\overline{\mathbf{B}} \in \mathbb{R}^{T \times M}$ denote discrete orthonormal Fourier basis matrices evaluated at the sample outputs $\{y_n\}_{n=1}^N$ and the Riemann grid points $\{\overline{y}_t\}_{t=1}^T$, respectively. The decoding weights are then related to their frequency-domain representations via:

$$w_d(y_n) = (\mathbf{B}\,\boldsymbol{\omega}_d)_n, \text{ and } w_d(\overline{y}_t) = (\overline{\mathbf{B}}\,\boldsymbol{\omega}_d)_t, \qquad (7)$$

where $\boldsymbol{\omega}_d = [\omega_{1,d}, \omega_{2,d}, \cdots, \omega_{M,d}]^\top$ are Fourier coefficients for feature $d$, drawn independently as:

$$\omega_{m,d} \sim \mathcal{N}(0, k_{m,d}), \quad k_{m,d} = \tilde{\rho}_d \exp\left(-\frac{1}{2} f_m^2 \ell_d^2\right) \text{ for } m = 1, \cdots, M, \qquad (8)$$

with $\tilde{\rho}_d = \sqrt{2\pi}\rho_d\ell_d$, and $f_m$ denoting the $m^{\text{th}}$ Fourier frequency. By truncating to $M \ll T, N$ basis functions, we drastically reduce the dimensionality of inference. In this formulation, inference reduces to estimating low-dimensional Fourier coefficients $\{\boldsymbol{\omega}_d\}_{d=1}^D$, which parameterize the decoding weight functions via a fixed orthonormal basis. This reparameterization turns the original infinite-dimensional inference problem into a tractable finite-dimensional one in the frequency domain.

## 3.3 Stochastic variational optimization in the frequency domain

We assume the variational distribution over frequency-domain weights is fully factorized across features and frequencies, with each component following a univariate Gaussian:

$$q(\boldsymbol{\omega}) = \prod_{d=1}^D \prod_{m=1}^M q(\omega_{m,d}), \quad q(\omega_{m,d}) \sim \mathcal{N}\left(\mu_{m,d}, \sigma_{m,d}^2\right). \qquad (9)$$

Under this assumption, the expected log-likelihood term in the ELBO (Eq. 5) becomes:

$$\mathbb{E}_{q_\psi}\left[\log p_\theta(\{\mathbf{x}_n\}_{n=1}^N \mid \boldsymbol{\omega})\right] = \sum_{n=1}^N \sum_{d=1}^D (\mathbf{B}\,\boldsymbol{\mu}_d)_n \, x_{n,d} - \mathbb{E}_{q_\psi}\left[\sum_{n=1}^N \log\left(\Delta \sum_{t=1}^T \exp\left(\sum_{d=1}^D (\overline{\mathbf{B}}\,\boldsymbol{\omega}_d)_t \, x_{n,d}\right)\right)\right],$$

where the second expectation is approximated via Monte Carlo sampling (Robert et al., 1999). The KL divergence term in the ELBO (Eq. 5), due to the Gaussian assumptions, admits a closed-form:

$$D_{KL}\left(q_\psi(\boldsymbol{\omega}) \| p_\theta(\boldsymbol{\omega})\right) = \frac{1}{2} \sum_d \left(\sum_m \left(\log\left(k_{m,d}/\sigma_{m,d}^2\right)\right) + \sum_m \sigma_{m,d}^2/k_{m,d} + \sum_m \left(\mu_{m,d}\right)^2/k_{m,d}\right).$$

We jointly optimize the variational parameters $\{\mu_{m,d}, \sigma_{m,d}\}_{m,d=1}^{M,D}$ and GP hyperparameters $\theta = \{\rho_d, \ell_d\}_{d=1}^D$ using stochastic variational inference with mini-batches of size $N' \ll N$, thus facilitating the implementation to scale with $N$ (Hoffman et al., 2013). We used the Adam optimizer (Kingma & Ba, 2017) for training, with all scale parameters (including $\theta$) optimized in log-space to ensure positivity. Extensive simulations (Appendix A.1) show that the proposed inference procedure reliably recovers both the decoding weights and hyperparameters across diverse settings.

**Computational scaling:** We evaluated the computational efficiency of CMLR across a range of settings for the number of neurons/features $D$, samples $N$, and Fourier components $M$, as shown in Appendix A.3, Fig. S3. Training time scales linearly with the number of neurons $D$: for example, convergence requires roughly $10^2$ seconds for $D \approx 200$ and $10^3$ seconds for $D \approx 2000$ on a standard laptop (Intel i7 2.4 GHz CPU, 16 GB RAM). Thanks to stochastic variational inference, training time increases only modestly with the number of samples $N$. In addition, both runtime and decoding performance are largely insensitive to the number of Fourier components $M$, indicating that relatively small $M$ values are sufficient for accurate estimation. Overall, Fig. S3 shows that CMLR is computationally efficient and robust across a broad range of dataset sizes and configurations.

## 3.4 Extension to Multi-Dimensional Outputs

While previous sections focused on scalar outputs, CMLR naturally extends to multi-dimensional settings, enabling multivariate conditional density estimation. For example, suppose the output

variable is two-dimensional, $\boldsymbol{y} = [y^{(1)}, y^{(2)}]$. We model the GP prior covariance of the decoding weight functions using an anisotropic RBF kernel (Rasmussen & Williams, 2006), which assigns separate length scales to each output dimension:

$$K_d\left(\boldsymbol{y}_{n'}, \boldsymbol{y}_{n''}\right) = \rho_d \exp\left(-\frac{1}{2}\left(\left(y_{n'}^{(1)} - y_{n''}^{(1)}\right)^2 \Big/ \left(\ell_d^{(1)}\right)^2 + \left(y_{n'}^{(2)} - y_{n''}^{(2)}\right)^2 \Big/ \left(\ell_d^{(2)}\right)^2\right)\right). \quad (10)$$

This results in the following modified frequency-domain representation of the prior variance (Eq. 8):

$$k_{m,d} = \tilde{\rho}_d \exp\left(-\frac{1}{2}\left(\left(f_m^{(1)}\right)^2 \left(\ell_d^{(1)}\right)^2 + \left(f_m^{(2)}\right)^2 \left(\ell_d^{(2)}\right)^2\right)\right) \text{ for } m = 1, \cdots, M, \quad (11)$$

where $\tilde{\rho}_d = 2\pi\rho_d \ell_d^{(1)} \ell_d^{(2)}$, and $f_m^{(1)}, f_m^{(2)}$ denote the Fourier frequencies in the two output dimensions. Our stochastic variational inference framework generalizes naturally to this multi-dimensional setting. We jointly estimate the variational parameters $\{\mu_{m,d}, \sigma_{m,d}\}_{m,d=1}^{M,D}$ along with the GP hyperparameters $\theta = \left\{\rho_d, \ell_d^{(1)}, \ell_d^{(2)}\right\}_{d=1}^{D}$. Fig. S4 presents simulation results demonstrating that the proposed inference procedure accurately recovers the true underlying parameters in the 2D stimulus setting.

## 4  OUTPUT PREDICTION USING THE TRAINED CMLR MODEL

Once the CMLR model is trained, the learned parameters can be used to predict outputs for test data at any desired resolution. For a target resolution $\delta$, we uniformly partition the output range $[y_{\mathsf{min}}, y_{\mathsf{max}}]$ into a total of $J$ classes, where $J = \mathrm{ceil}\left((y_{\mathsf{max}} - y_{\mathsf{min}})/\delta\right)$. Let $\tilde{y}_j$ denote the center of the $j^{\mathrm{th}}$ bin. Based on this uniform grid, we construct the corresponding decoding Fourier basis matrix $\mathbf{B}^{\mathsf{dec}} \in \mathbb{R}^{J \times M}$, and compute the decoding weights for features $d = 1, \cdots, D$ at resolution $\delta$ as:

$$w_d(\tilde{y}_j) = \left(\mathbf{B}^{\mathsf{dec}} \times \boldsymbol{\mu}_d\right)_j.$$

Given a test sample $\mathbf{x}_n$, we then compute the posterior over the output grid via the softmax function:

$$p\left(y_n = \tilde{y}_j \mid \mathbf{x}_n, \mathbf{w}(y)\right) = \frac{\exp\left(\mathbf{w}(\tilde{y}_j)^{\top}\mathbf{x}_n\right)}{\sum_{j'=1}^{J} \exp\left(\mathbf{w}(\tilde{y}_{j'})^{\top}\mathbf{x}_n\right)}.$$

This yields the full posterior conditional distribution. We then decode the output using either the posterior mean or the posterior mode, depending on the task objective. The posterior mean yields smooth, high-resolution predictions:

$$\widehat{y}_{\mathsf{mean}} = \sum_{j=1}^{J} \tilde{y}_j \cdot p\left(y_n = \tilde{y}_j \mid \mathbf{x}_n, \mathbf{w}(y)\right),$$

while the posterior mode selects the most likely output:

$$\widehat{y}_{\mathsf{mode}} = \arg\max_{j \in \{1:J\}} p\left(y_n = \tilde{y}_j \mid \mathbf{x}_n, \mathbf{w}(y)\right).$$

The mean decoder is particularly well-suited for regression-style tasks, whereas the mode decoder is preferable for minimizing classification error. Note that we can set $\delta$ to be arbitrarily small to achieve a desired level of resolution in output space.

## 5  APPLICATIONS TO NEURAL DECODING

We evaluated CMLR on four real-world neural datasets spanning diverse brain regions and species: mouse V1 (Stringer et al., 2021), monkey V1 (Graf et al., 2011), mouse hippocampal CA1 (Hazon et al., 2022; Jercog & Schnitzer, 2022), and monkey motor cortex (Glaser et al., 2018; 2020). For each dataset, we performed 5-fold cross-validation. In each fold, the data were split into $80\%$ training and $20\%$ test sets. For DNN and XGBoost, we further reserved $20\%$ of the training set (i.e., $16\%$ of the total data) as a validation set for hyperparameter tuning via Bayesian optimization (Gardner et al., 2014). For CMLR and the other baselines, hyperparameters were fixed per dataset. The selected CMLR design parameters and their rationale are provided in Appendix A.5. Final performance was averaged over the test sets across all folds. We compared CMLR with the following four baselines.

**Naive Bayes (NB)**: We implemented a continuous-output variant of the Naive Bayes decoder that shares the same Gaussian process (GP) priors over decoding weight functions $w_d(y)$ as CMLR, using RBF kernels and Fourier-domain inference. However, unlike CMLR, NB assumes conditional independence of neural responses across features given the output, yielding the following likelihood:

$$p\left(\{\mathbf{x}_n\}_{n=1}^N \mid \mathbf{w}(y)\right) = \prod_{n=1}^N \prod_{d=1}^D p(x_{n,d}|w_d(y)).$$

We used Gaussian observation models for calcium imaging data and Poisson observation models for spike count data, following Greenidge et al. (2024). This formulation preserves the functional interpretability of weights as tuning curves and supports CDE, but ignores correlations across neurons.

**Flexible nonparametric conditional density estimation (FlexCode)**: FlexCode is a state-of-the-art nonparametric CDE method that reformulates CDE as a series expansion problem, estimating the basis coefficients via regression (Izbicki & Lee, 2017). We use the publicly available implementation from Izbicki & Lee (2017) with Random Forest regression to estimate coefficients.

**Extreme Gradient Boosting (XGBoost)**: XGBoost is a widely used implementation of gradient-boosted decision trees (Chen & Guestrin, 2016). We used the implementation from Glaser et al. (2020), which was specifically tuned for neural decoding and used multiple adjacent time bins as input. Similar to Glaser et al. (2020), we optimized hyperparameters using Bayesian optimization (Gardner et al., 2014) based on validation-set $R^2$, searching over tree depth, number of trees, and learning rate.

**Deep neural network (DNN)**: DNNs consist of multiple layers of nonlinear transformations that map inputs to outputs (Orbach, 1962; Goodfellow et al., 2016). We used the architecture and training procedure from Glaser et al. (2020), which was specifically designed for neural decoding. Following this work, we tuned the number of hidden units, dropout rate, and number of epochs using Bayesian optimization based on validation $R^2$, and we used multiple adjacent time bins as input.

## 5.1 Primary visual cortex (V1): decoding drifting grating orientation

We first applied our method to three two-photon calcium imaging datasets recorded from mouse primary visual cortex during drifting grating stimuli (Stringer et al., 2021). Each dataset included $D$=11311–20616 neurons (input features) and $N = 4282$–4469 trials, with stimulus orientations (outputs) uniformly sampled from $[0, 2\pi]$. We used the same features $\mathbf{x}$ as in Stringer et al. (2021).

Fig. 2 summarizes the results. Fig. 2A compares decoding weights from the CMLR and Naive Bayes models for selected neurons, normalized to $[0, 1]$ in amplitude. CMLR weights appear smoother, consistent with the larger inferred length scales shown in Fig. 2B. We next assessed mean absolute circular error across decoding grid resolutions, controlled by the number of decoding classes $J$ (Fig. 2C). CMLR consistently outperformed FlexCode and Naive Bayes, with performance gains saturating beyond $J \approx 5000$. We also compared CMLR to XGBoost and DNN decoders (Fig. 2D–E). These models are omitted from Fig. 2C because point-estimate regressors lack a conditional density and do not naturally admit resolution-dependent analysis. Fig. 2D shows decoded versus true orientation, with CMLR predictions clustering tightly around the identity line, with most large errors occurring near $180°$, reflecting underlying bimodality. Fig. 2E shows that CMLR achieves the lowest decoding error (mean $\pm$ SD: $3.1 \pm 9.3°$, median: $2.1°$, Inter-Quartile range (IQR): $2.7°$), followed by FlexCode ($3.2 \pm 5.5°$, $2.2°$, $2.8°$), Naive Bayes ($4.9 \pm 10.8°$, $3.3°$, $4.4°$), XGBoost ($13.6 \pm 23.4°$, $6.9°$, $11.1°$), and DNN ($18.3 \pm 23.6°$, $11.9°$, $16.5°$). Moreover, the scalability analysis in Appendix A.6 (Fig. S5) shows that CMLR experiences only modest reductions in accuracy as $D$ and $N$ decrease, with an even larger performance gap over XGBoost and DNN in low-data settings.

Next, we applied our method to five electrophysiological datasets from monkey primary visual cortex, recorded under drifting grating stimuli (Graf et al., 2011). Each dataset contained spiking activity from $D = 113$–148 neurons across 72 discrete grating orientations (spaced at $5°$ intervals), with a total of $N = 3600$ trials. Although CMLR is designed for continuous outputs, it naturally accommodates discrete settings without modification. Using the same inference settings as before, we found that CMLR and FlexCode consistently outperformed all baseline models (Fig. S6).

As a correlation-aware decoder, CMLR stands in contrast to the correlation-blind Naive Bayes model (Wei et al., 2024; Greenidge et al., 2024), reinforcing the importance of modeling noise correlations for accurate decoding in V1. It is also noteworthy that the discrete MLR model in Greenidge et al.

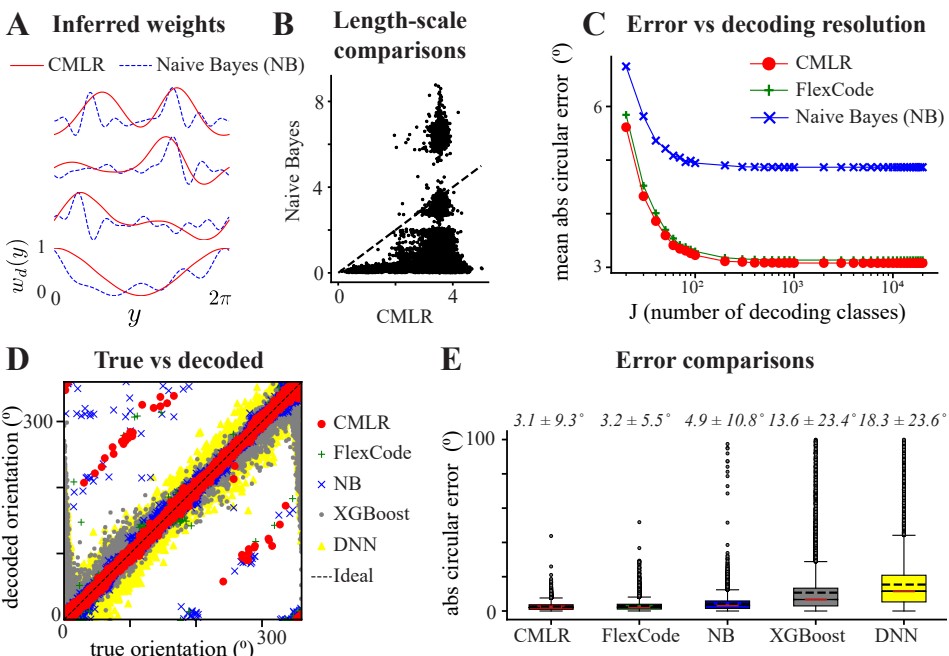

Figure 2: Application of the proposed method to mouse V1 data (data from Stringer et al. (2021)). (A) Decoding weights inferred by the CMLR and Naive Bayes (NB) models for selected neurons, normalized to $[0, 1]$. (B) Scatter plots comparing inferred length scales by the CMLR model versus NB model for individual neurons. (C) Mean absolute circular error as a function of the number of decoding classes $J$ for CMLR, FlexCode, and NB. (D) Scatter plots of decoded versus true stimulus orientations across five models: CMLR, FlexCode, NB, XGBoost, and DNN. (E) Box plots comparing the absolute circular decoding error across all methods. Mean errors $\pm$ standard deviation are indicated as insets.

(2024) can be seen as a special case of CMLR in the low-$J$ regime. As $J$ increases, CMLR enables principled evaluation of resolution-dependent performance in the high-resolution limit, rather than relying on arbitrary discretization. This is especially relevant for continuous or circular variables, such as orientation, where fine-grained distinctions are behaviorally meaningful. The learned decoding functions $w_d(y)$ act like tuning curves, providing smooth and interpretable weights that link individual neurons to specific stimuli or behavioral outputs. This makes CMLR not only a powerful decoder but also a useful tool for probing population coding in V1.

## 5.2 HIPPOCAMPUS CA1: DECODING POSITION DURING SPATIAL NAVIGATION

We next applied CMLR to eight calcium imaging datasets recorded from pyramidal neurons in the mouse hippocampus CA1 region, while the animals navigated a 120-cm linear track to collect water rewards (data from Hazon et al. (2022); Jercog & Schnitzer (2022)). Each dataset contained $D = 151$–$497$ neurons (input features), and we used the same preprocessed neural features $\mathbf{x}$ as in Hazon et al. (2022); Jercog & Schnitzer (2022), downsampled by a factor of 10, yielding $N = 3600$–$5524$ samples. Position outputs were normalized to $[0, 1]$ for ease of comparison.

Fig. 3 summarizes the results. Panel A shows that CMLR consistently outperformed FlexCode and Naive Bayes in mean absolute error across decoding grid resolutions. This reinforces the benefit of modeling population-level structure in hippocampal circuits. Panel B displays scatter plots of true versus decoded positions for all models, with CMLR predictions aligning most closely with the identity line. Panel C compares Euclidean decoding errors: CMLR achieved the lowest error (mean $\pm$ SD: $0.15 \pm 0.31$, median: $0.01$, IQR: $0.09$), followed by FlexCode ($0.16 \pm 0.30$, $0.01$, $0.07$), Naive Bayes ($0.16 \pm 0.31$, $0.02$, $0.06$), XGBoost ($0.16 \pm 0.13$, $0.11$, $0.16$), and DNN ($0.18 \pm 0.16$, $0.15$, $0.19$). Together, these findings demonstrate the flexibility and robustness of CMLR across continuous decoding tasks in hippocampal circuits.

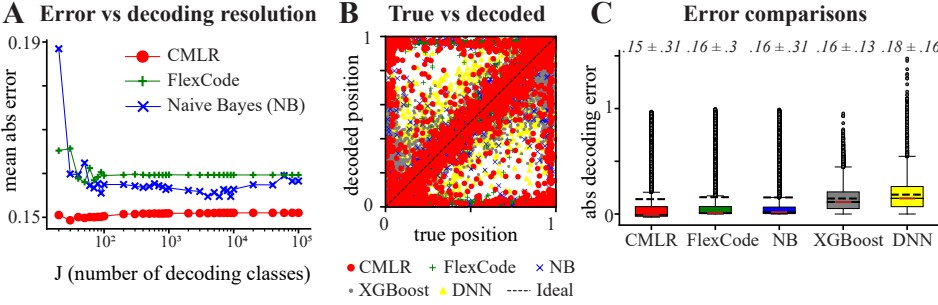

Figure 3: Application of the proposed method to mouse hippocampal data (Hazon et al., 2022; Jercog & Schnitzer, 2022). (A) Mean absolute error for CMLR, FlexCode, and Naive Bayes (NB) as a function of the number of decoding classes $J$. (B) Scatter plots comparing true versus decoded spatial positions for CMLR, FlexCode, NB, XGBoost, and DNN. (C) Box plots comparing absolute decoding errors across models (insets: mean $\pm$ standard deviation).

## 5.3 MOTOR CORTEX: DECODING 2D VELOCITY IN A REACHING TASK

We then applied CMLR to a dataset recorded from the motor cortex of a monkey performing a target-reaching task (data from Glaser et al. (2018; 2020)). In this task, the monkey controlled a cursor on a screen using a manipulandum (Glaser et al., 2018), and our goal was to decode the cursor's two-dimensional (2D) velocity in the $x$ and $y$ directions. We used the same preprocessed neural features $\mathbf{x}$ as in Glaser et al. (2020). The dataset comprised $N = 25{,}299$ samples over a 21-minute session, with spiking activity from $D = 164$ neurons.

Fig. 4 summarizes the results. Panel A shows inferred decoding weights for seven example neurons, revealing diverse spatial tuning profiles in the 2D velocity space, with all neurons shown in Fig. S7. Additional analyses in Fig. S8 show that the inferred weight functions reflect empirical spike–velocity structure and remain smooth and coherent even when trained on restricted portions of the velocity space. Panel B compares true and predicted velocities for all models; CMLR aligns closely with the identity line. Panel C presents Euclidean decoding errors and reports the coefficient of determination ($R^2$): CMLR achieves strong performance (0.53), outperforming FlexCode (0.35) and Naive Bayes ($-0.43$), and approaching XGBoost (0.55) and DNN (0.58). The higher scores of XGBoost and DNN are expected, given the large size of this dataset, which favors high-capacity nonlinear models. CMLR

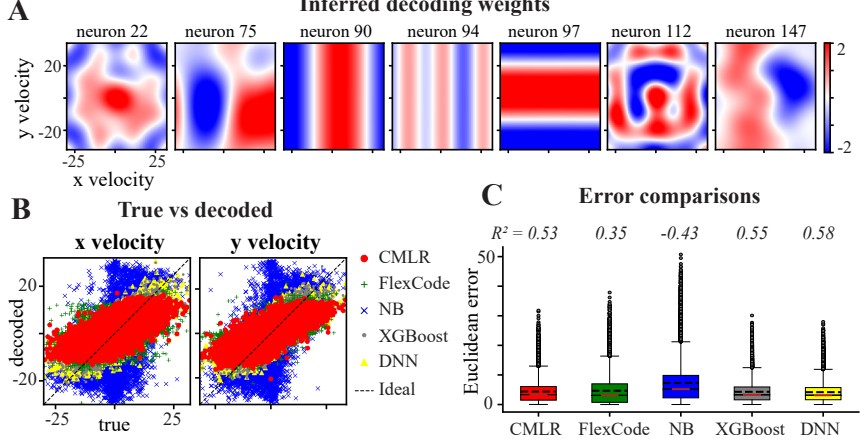

Figure 4: Decoding 2D cursor velocity from motor cortex data (Glaser et al., 2018; 2020). (A) Decoding weights inferred by CMLR for seven representative neurons (full set shown in Fig. S7) (B) Scatter plots comparing true versus decoded $x$ (left) and $y$ (right) cursor velocities for all methods: CMLR, FlexCode, Naive Bayes (NB), XGBoost, and deep neural networks (DNN). (C) Box plots of Euclidean decoding error across all methods (insets report the $R^2$ score).

nevertheless remains competitive while also providing full conditional densities and interpretable tuning functions, capabilities that are not available in point-estimate regressors. Overall, these findings demonstrate that CMLR extends naturally to multivariate-output CDE and continuous decoding.

## 5.4 Calibration of Predictive Uncertainty

Finally, we evaluated the calibration of the predictive posteriors from the two CDE methods, CMLR and FlexCode, using PIT histograms and quantile calibration curves (Fig. S9; see Appendix A.10 for details). Across mouse V1, macaque V1, and mouse hippocampal CA1, CMLR consistently produces well-calibrated posteriors: PIT histograms are close to uniform, quantile calibration curves track the identity line, and PIT values vary smoothly with decoding error. FlexCode, in contrast, shows systematic miscalibration, with peaked or multimodal PIT distributions, under-coverage in quantile calibration, and irregular PIT–error relationships. These results indicate that CMLR provides more reliable and interpretable uncertainty estimates, which is important in neural decoding applications where posterior uncertainty guides scientific interpretation and downstream decisions, such as identifying ambiguous stimuli or estimating decoding confidence (Wei et al., 2024).

## 5.5 Run time comparisons

Table 1 reports average training times for all methods on a 2.4 GHz Intel i7 CPU with 16 GB RAM (all without parallelization). CMLR trains within minutes to a few hours, depending on dataset size, with run times comparable to FlexCode and faster than Naive Bayes. XGBoost and DNN are generally faster, but they do not provide full conditional densities or calibrated uncertainty. Overall, CMLR achieves competitive computational efficiency while offering richer probabilistic outputs.

Table 1: Run time comparisons for all methods. Values are averaged over folds.

| Dataset | $D$ (# neurons) | $N$ (# trials) | CMLR | FlexCode | NB | XGBoost | DNN |
|---------|-----------------|----------------|------|----------|-----|---------|------|
| Mouse V1 | 11–21 k | 4.3–4.5 k | 3.3 h | 2.3 h | 22 h | 4.4 h | 33 m |
| Monkey V1 | 113–148 | 3.6 k | 5.5 m | 25 s | 2.6 m | 2 m | 1.9 m |
| Mouse CA1 | 151–497 | 3.6–5.5 k | 10 m | 47 s | 4 m | 4 m | 2 m |
| Monkey MC | 164 | 25 k | 10 m | 11 m | 30 m | 23 s | 5 m |

## 6 Conclusion and Discussion

We introduce the Continuous Multinomial Logistic Regression (CMLR) model, a novel exponential-family framework for scalable, nonparametric conditional density estimation (CDE). CMLR extends multinomial logistic regression (MLR) to continuous outputs by replacing discrete class weights with smooth, output-indexed functions drawn from Gaussian process (GP) priors, yielding normalized densities while preserving MLR's additive, interpretable structure. For efficient training, we develop a memory-efficient stochastic variational inference algorithm in the Fourier domain, leveraging GP stationarity for kernel diagonalization and basis truncation. Applied to large-scale neural datasets from mouse and monkey V1, hippocampus CA1, and motor cortex, CMLR outperforms Naive Bayes, XGBoost, deep neural networks, and the leading CDE method FlexCode, while providing calibrated posteriors and interpretable structure. Comparisons with Naive Bayes demonstrate that explicitly modeling correlation structure substantially improves decoding performance across all brain areas.

To our knowledge, CMLR represents the first application of CDE to neural decoding, enabling flexible estimation of full posterior distributions over behaviorally or perceptually relevant variables. While modern high-capacity nonlinear models may achieve stronger predictive accuracy in very large-data regimes, they typically require extensive hyperparameter tuning and offer limited interpretability. CMLR plays a complementary role: it yields fully probabilistic and well-calibrated conditional densities, provides access to the full posterior rather than point estimates, is data-efficient with stable performance across datasets, and remains easy to train because it requires only a few design parameters and no dataset-specific hyperparameter tuning. In addition, CMLR produces transparent tuning functions that can be directly visualized and compared across neurons and conditions, in the same manner as classical systems-neuroscience tuning curves, a property not typically supported by black-box models. We therefore view CMLR as a practical and interpretable baseline or diagnostic model that complements more complex nonlinear approaches in modern neural decoding settings.

**Connections to Prior Work:** CMLR builds on a rich body of research in nonparametric Bayesian inference using Gaussian processes (Williams & Barber, 1998; Girolami & Rogers, 2006; Bishop, 2006), but is conceptually distinct from standard GP regression (Rasmussen & Williams, 2006; Chan, 2013) and GP classification (Liu et al., 2022). In GP regression, the output is modeled as a GP function of the input, yielding Gaussian predictive distributions. In GP classification, a latent GP is passed through a nonlinear link to produce class probabilities over discrete outputs. Both approaches lack the flexibility to represent rich or structured conditional densities over continuous outputs, and related methods that approximate uncertainty for point estimators, such as Laplace approximation (Daxberger et al., 2022), also do not provide full conditional densities. In contrast, CMLR places GP priors on weight functions defined over the output space, enabling flexible estimation of full conditional densities. CMLR is also closely related to logistic Gaussian processes (Tokdar et al., 2004) and early unconditional density models such as the Gaussian process density sampler (Murray et al., 2008), but generalizes them into a conditional, feature-decomposed framework that preserves the additive structure and interpretability of MLR. As a CDE model, CMLR complements several existing approaches, including kernel-ratio estimators (Bashtannyk & Hyndman, 2001; Holmes et al., 2012; Sugiyama et al., 2010), mixture density networks (Bishop, 1994), neural-kernel mixture models (Ambrogioni et al., 2017), conditional normalizing flows (Papamakarios et al., 2017), logistic Gaussian process partition models (Payne et al., 2019), Lindsey's Method (Gao & Hastie, 2022), and histogram trees (Yang & van Leeuwen, 2024). However, many of these methods face limitations in scalability, interpretability, or statistical robustness in high-dimensional settings. CMLR offers a nonparametric and additive alternative that provides interpretable weight functions, supports multiple output dimensions, and achieves computational scalability through structured inference. CMLR also has conceptual links to neural operator models, particularly those defined over continuous input–output mappings (Kovachki et al., 2023; Li et al., 2021). To scale CMLR to large datasets, we leverage sparse Fourier-domain representations developed for variational GP inference (Hensman et al., 2018; Keeley et al., 2020; Gondur et al., 2024), closely related in motivation and formulation to Fourier Neural Operators (Li et al., 2021). Our model also builds directly on the discrete multinomial logistic regression framework for neural decoding in Greenidge et al. (2024), extending it to handle continuous outputs in a principled and computationally efficient manner.

**Limitations and Future Directions:** While CMLR provides a unified and scalable framework for CDE, several directions remain open for future work. First, unlike XGBoost and DNNs that can implicitly capture temporal continuity by incorporating activity from multiple adjacent time bins (Glaser et al., 2020), CMLR does not yet include explicit output-space priors, such as temporal or spatial smoothness, which could be useful in navigation or motor decoding. Integrating structured priors into the conditional likelihood, or combining CMLR with latent dynamical models (Park et al., 2015; Damianou et al., 2011), may enhance temporal generalization. Second, the current CMLR model treats each decoding weight independently; multivariate GP priors (Bonilla et al., 2007; Keeley et al., 2020) could share structure across neurons or time and potentially improve generalization. However, such coupling would obscure neuron-specific tuning curves and thus compromise interpretability. Third, while fixed Fourier-domain bases and RBF kernels enable scalable inference, more flexible alternatives such as adaptive basis functions (Evans & Nair, 2018) or spectral mixture kernels (Paciorek & Schervish, 2003) could further enhance accuracy, though it remains to be tested whether such extensions retain the same computational scalability. Fourth, while additivity over inputs ensures scalability, convexity, and interpretability, incorporating low-rank or kernelized interaction models (Duvenaud et al., 2011) could enhance model expressiveness by capturing higher-order feature dependencies. Fifth, although we used Riemann integration to support tractable variational inference, advanced numerical techniques such as general quadrature methods (Hildebrand, 1987), polynomial approximations (Trefethen, 2019), or adaptive binning (Wand, 1997) may offer improvements in accuracy or efficiency. Sixth, hybrid approaches that combine inducing-point methods with Fourier domain representations (Hensman et al., 2018) could further improve computational scalability. Finally, although we focused here on sensory and motor decoding, CMLR is broadly applicable to other neural decoding tasks, including decoding motor intention (Tam et al., 2019), speech (Chen et al., 2024), spatial attention (Smith et al., 2013), and decision variables (Baeg et al., 2003). Beyond neuroscience, CMLR may also benefit other CDE applications such as head pose estimation (Murphy-Chutorian & Trivedi, 2009), time-to-event modeling (Gensheimer & Narasimhan, 2019), climate prediction (Rasp et al., 2018), photometric redshift estimation (Dalmasso et al., 2020), and geolocation and trajectory forecasting (Rhinehart et al., 2019). Together, these directions position CMLR as a versatile CDE framework for neural decoding and beyond.

ACKNOWLEDGMENTS

This work was supported by the NIH T32 institutional training grant (T32MH065214), the Simons Collaboration on the Global Brain (SCGB AWD543027), the NIH BRAIN Initiative (R01DA056404, R01EB02694), the National Eye Institute (NEI) of the National Institutes of Health (NIH) (R01EY033064), and a U19 NIH-NINDS BRAIN Initiative Award (5U19NS123716). We also thank C. Stringer, M. Michaelos, D. Tsyboulski, S. Lindo, and M. Pachitariu for providing the publicly available mouse V1 datasets; A. B. A. Graf, A. Kohn, M. Jazayeri, and J. A. Movshon for providing the monkey V1 datasets; P. Jercog and M. Schnitzer for providing the publicly available mouse hippocampus CA1 datasets; and J. I. Glaser, M. G. Perich, P. Ramkumar, L. E. Miller, and K. P. Kording for providing the publicly available monkey motor cortex dataset.

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

# A  APPENDIX

This appendix provides supplementary material that supports, extends, and validates the core contributions presented in the main text.

- Section A.1 presents the simulation study, demonstrating that the proposed inference procedure robustly recovers the decoding weight functions and hyperparameters.

- Section A.2 (Fig. S2) presents full decoding weight visualizations from the simulation study in Section A.1.

- Section A.3 (Fig. S3) reports additional simulation results evaluating inference performance under varying model dimensions, including the number of features ($D$), sample size ($N$), and Fourier resolution ($M$), complementing the analysis in Section A.1.

- Section A.4 (Fig. S4) extends the simulation-based validation to two-dimensional output spaces, demonstrating accurate recovery of decoding weights and GP hyperparameters in a multidimensional setting.

- Section A.5 describes the CMLR design parameters, provides guidance on how to select them, and Table 2 summarizes the settings used in each real-data application.

- Section A.6 presents a scalability analysis of CMLR with respect to neuron count and dataset size on the mouse V1 dataset, and Fig. S5 compares its performance and training time with XGBoost and DNN.

- Section A.7 (Fig. S6) compares CMLR performance with other methods on the monkey V1 dataset recorded under drifting grating stimuli (data from Graf et al. (2011)), extending the results in Section 5.1.

- Section A.8 (Fig. S7) provides a complete visualization of decoding weight functions inferred from the monkey motor cortex dataset (extending Fig. 4), highlighting a rich diversity of smooth and interpretable 2D tuning profiles across all recorded neurons.

- Section A.9 (Fig. S8) examines the interpretability and generalization of the inferred decoding weights by comparing CMLR weight functions to empirical spike–velocity maps and by analyzing how the inferred functions behave when trained on progressively restricted subsets of the velocity space for the monkey motor cortex dataset (extending Fig. 4).

- Section A.10 details the procedures used to assess calibration of CMLR and FlexCode posterior densities, including PIT histograms and quantile calibration curves, with results evaluated across datasets (Fig. S9).

- Section A.11 declares the limited use of large language models (LLMs) for minor editing of the manuscript.

These results collectively reinforce the accuracy, scalability, and interpretability of the CMLR framework.

A.1 SIMULATION STUDY

To assess the accuracy and robustness of our inference framework, we conducted simulations using synthetic data. We considered a continuous output space of orientation angles $\theta \in [0, 2\pi)$ with $N = 4000$ samples. The input consisted of $D = 200$ features, with decoding weights $w_d(y)$ drawn from Gaussian process (GP) priors. Each weight function was sampled on a fine grid of $J$ points, $\{y_{\text{grid}}^{(j)}\}_{j=1}^{J}$, with GP hyperparameters drawn as $\ell_d \in [0.1, 1.5]$ (length scales) and $\rho_d \in [0.5, 2.5]$ (variances). Feature vectors $\mathbf{x}_n$ were sampled independently from a standard Gaussian distribution. Outputs $y_n \in \{y_{\text{grid}}^{(j)}\}_{j=1}^{J}$ were then drawn from the conditional density:

$$p\Big(y = y_{\text{grid}}^{(j)} \mid \mathbf{x}_n, \mathbf{w}(y)\Big) = \frac{\exp\Big(\mathbf{w}(y_{\text{grid}}^{(j)})^\top \mathbf{x}_n\Big)}{\sum_{j'=1}^{J} \exp\Big(\mathbf{w}(y_{\text{grid}}^{(j')})^\top \mathbf{x}_n\Big)},$$

using a finite grid approximation. Algorithm 1 summarizes the procedure. Note that this procedure can be made arbitrarily accurate by refining the output grid, and exact sampling is also possible via inverse CDF sampling due to the smoothness of the GP-drawn weight functions.

---

**Algorithm 1** Simulation procedure for generating synthetic data

---

1: Define an output grid $\{y_{\text{grid}}^{(j)}\}_{j=1}^{J}$ with $y_{\text{grid}}^{(j)} \in [0, 2\pi)$
2: **for** $d = 1$ to $D$ **do**
3:     Sample weight function $w_d(y)$ from a GP prior at $\{y_{\text{grid}}^{(j)}\}_{j=1}^{J}$ with $\ell_d \in [0.1, 1.5]$, $\rho_d \in [0.5, 2.5]$
4: **end for**
5: **for** $n = 1$ to $N$ **do**
6:     Sample feature vector $\mathbf{x}_n \sim \mathcal{N}(0, \mathbf{I}_D)$
7:     Compute posterior $p(y \mid \mathbf{x}_n, \mathbf{w}(y))$ as above
8:     Sample observed output $y_n \sim p(y \mid \mathbf{x}_n, \mathbf{w}(y))$
9: **end for**

---

For inference, we used $T = 100$ bins, $M = 17$ Fourier components, a mini-batch size of $N' = 1500$, learning rate 0.05, and 3 Monte Carlo samples to approximate the ELBO during training.

Fig. S1 summarizes the results. Panel A shows the ELBO trajectory over 580 iterations (104 seconds on a 2.4GHz Intel i7 CPU with 16GB RAM), exhibiting smooth, monotonic convergence and confirming stable stochastic optimization. Panel B compares ground truth and inferred GP hyperparameters across all $D = 200$ features. Inferred values closely track the identity line, with normalized absolute errors of $11.35\% \pm 12.14\%$ for length scales and $48.56\% \pm 54.81\%$ for variances. Larger errors arose for flatter functions (longer length scales or larger variances), where identifiability is inherently limited. Panel C shows true vs. inferred decoding weights for 5 representative features (full results in Fig. S2), with mean error across all features of $7.46\% \pm 5.11\%$, confirming recovery of both coarse- and fine-scale structure.

To further assess robustness, we performed additional simulations under varying conditions. Fig. S3 reports results as we varied the number of input features ($D$), number of samples ($N$), and number of Fourier components ($M$). Performance improves systematically with larger $N$ and $M$, while the method remains computationally scalable and statistically accurate in high-dimensional regimes. Notably, strong performance was achieved with as few as $M \approx 20$ Fourier components, supporting the use of compact frequency-domain representations to reduce dimensionality and prevent overfitting. Together, these findings demonstrate that our variational inference framework enables efficient and reliable recovery of both hyperparameters and decoding weights across diverse settings.

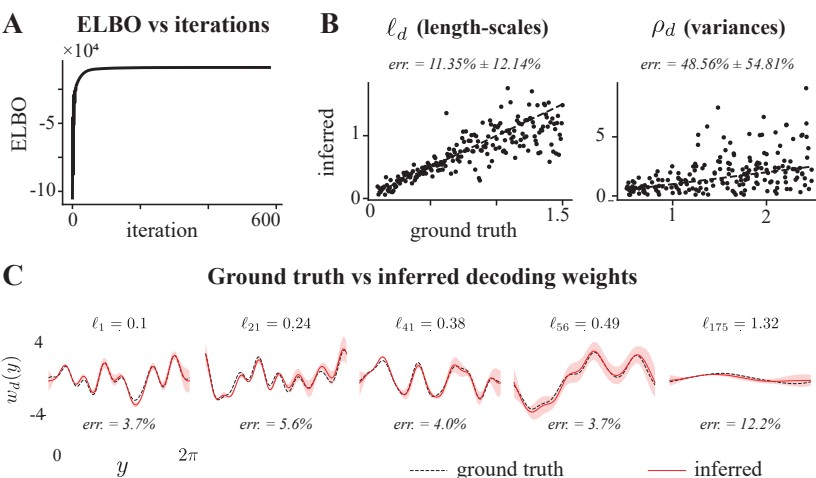

Figure S1: Simulation study results. (A) ELBO versus training iterations. (B) Scatter plots comparing ground truth versus inferred length scales (left) and variances (right) across features; insets report mean and standard deviation of normalized absolute errors. (C) Decoding weights for 5 representative features: ground truth (black) vs. inferred weights (red), shaded regions show posterior standard deviation. True length scales are shown above each subplot; normalized mean absolute errors are shown below. Full results in Fig. S2.

## A.2 COMPREHENSIVE VISUALIZATION OF TRUE VS INFERRED DECODING WEIGHTS (EXTENSION OF FIG. S1)

To provide a complete view of model performance, we present decoding weight recovery for all 200 simulated features used in the simulation study in Section A.1. This figure extends the subset shown in Fig. S1 and illustrates how well the inferred weights (red) match the ground truth (black) across all features. Features with varying ground truth GP length scales are included, demonstrating that our inference method successfully recovers both broad and sharply tuned weight functions.

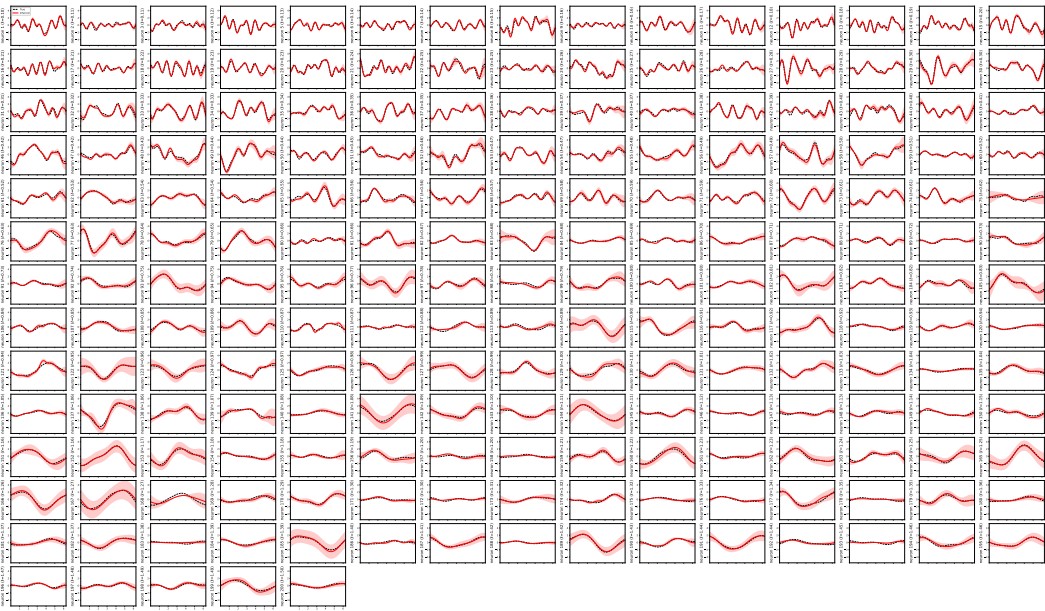

Figure S2: Full decoding weight recovery for all 200 simulated features, extending the results in Fig. S1. Each subplot shows the ground truth decoding weights (black) and the inferred weights (red) across the continuous output space for one feature, with shaded regions showing posterior standard deviation. The true GP length scale is indicated in parentheses. The results demonstrate that the proposed inference procedure accurately captures both smooth and localized tuning profiles.

### A.3   ROBUSTNESS AND SCALABILITY OF CMLR INFERENCE ACROSS SIMULATION SETTINGS (EXTENSION OF SECTION A.1)

To assess the robustness and computational efficiency of the CMLR model, we systematically evaluated its performance across a range of controlled simulation scenarios, extending the results in Section A.1. Fig. S3 summarizes how inference accuracy and training time vary with the number of input features ($D$), number of observed samples ($N$), and number of Fourier components ($M$). These results extend the main text in Section 3 and the simulation results in Fig. S1 by confirming that CMLR maintains high accuracy in hyperparameter and weight recovery while scaling gracefully with data size and model complexity.

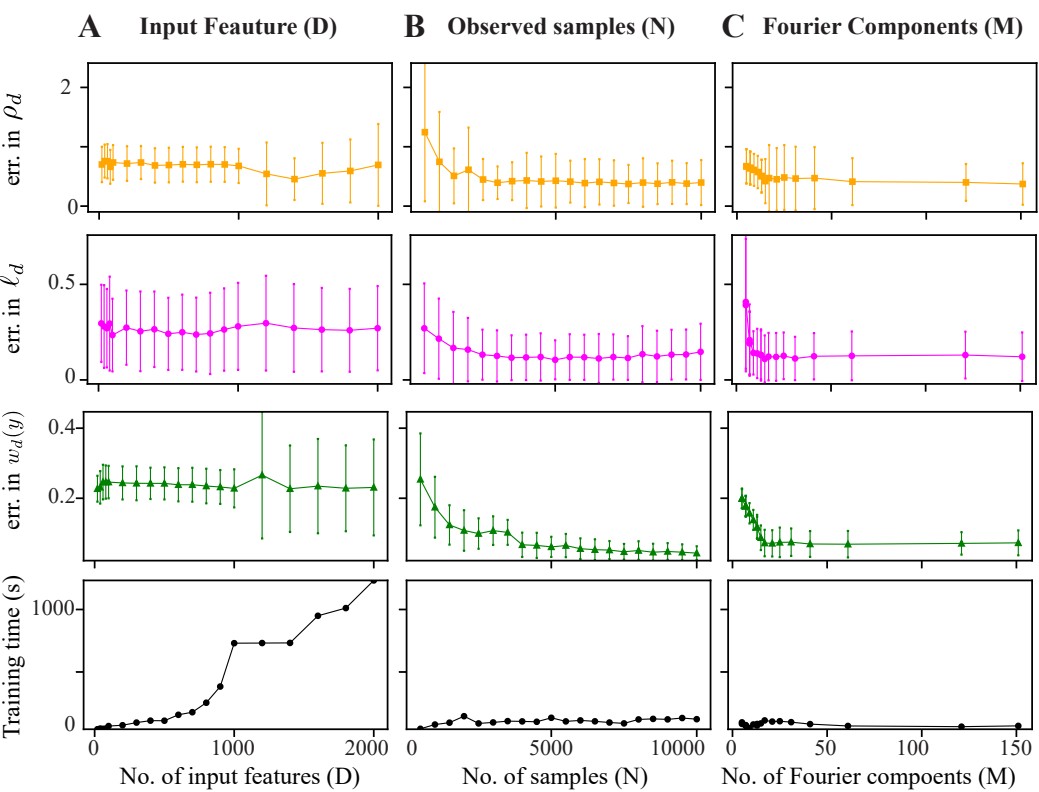

Figure S3: Simulation performance comparison under varying conditions. Rows (top to bottom) show: mean $\pm$ standard deviation of the normalized absolute error in inferred variances ($\rho_d$), length scales ($\ell_d$), and decoding weights ($w_d(y)$), followed by training time in seconds (Intel i7 2.4GHz CPU, 16GB RAM). (A) Varying the number of input features $D$ shows that inference accuracy remains stable across all metrics, with only modest increases in training time. (B) Increasing the number of observed samples $N$ substantially improves inference accuracy for all parameters, reflecting the benefit of additional data for GP hyperparameter and weight recovery. Notably, training time increases only marginally due to the use of stochastic variational inference. (C) Varying the number of Fourier components $M$ reveals that performance improves with increasing $M$, particularly in recovering fine-scale structure in decoding weights. However, the computational cost does not increase substantially due to the efficiency of the frequency-domain formulation. Notably, near-asymptotic performance is achieved with as few as $M \approx 20$ components. In practice, keeping $M$ small is advisable to reduce dimensionality and mitigate the risk of overfitting. These results demonstrate that CMLR offers reliable and scalable inference across a wide range of practical settings, with strong performance attainable using moderate data sizes and low-dimensional frequency representations.

## A.4 VALIDATION OF INFERENCE ACCURACY IN THE TWO-DIMENSIONAL SETTING (EXTENSION OF SECTION 3.4)

To confirm the accuracy and stability of CMLR in two-dimensional decoding tasks described in Section 3.4, we conducted a simulation study with synthetic neurons tuned to 2D outputs. Fig. S4 illustrates key results: convergence behavior of the variational inference algorithm, recovery of GP hyperparameters, and comparison of inferred vs. true decoding weights. Together, these analyses validate the model's ability to recover both the coarse and fine-grained structure of 2D tuning profiles, demonstrating the reliability of CMLR in multidimensional output spaces.

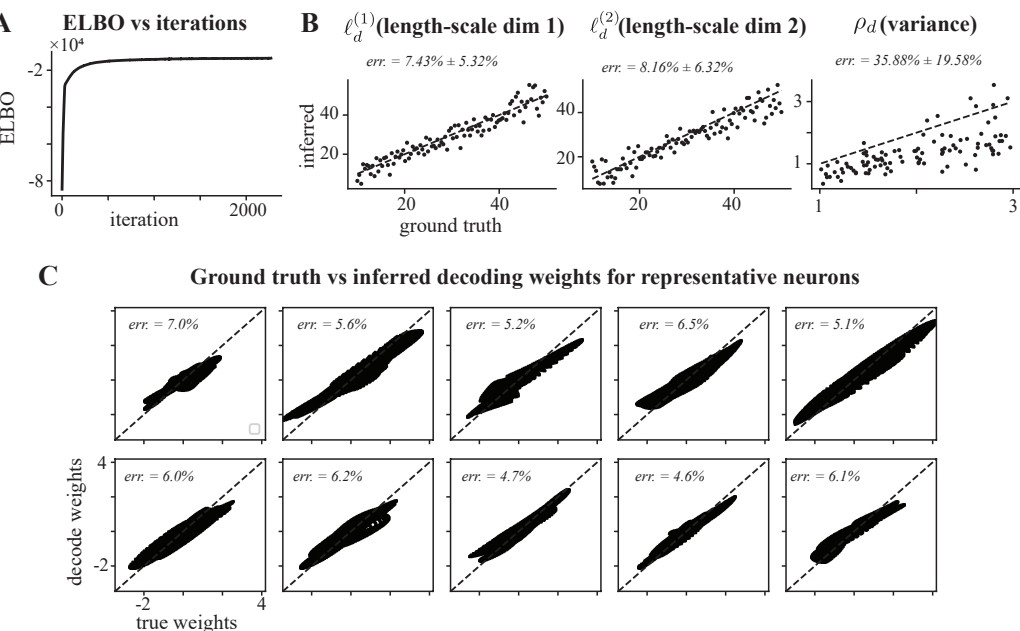

Figure S4: Simulation study validating inference accuracy in the 2D setting. (A) Evidence lower bound (ELBO) over training iterations, showing smooth and stable convergence of the variational inference algorithm. (completed in 469 seconds on a 2.4GHz Intel i7 CPU with 16GB RAM) (B) Scatter plots comparing ground truth and inferred GP hyperparameters (length scales and variances) across both output dimensions. Each point corresponds to one of the $D = 100$ simulated neurons; proximity to the identity line indicates accurate recovery. (C) Comparison of inferred and ground truth decoding weights for a subset of representative neurons. Each subplot shows a scatter plot of the true versus recovered weights, with the normalized mean absolute error shown as an inset. Points closely align with the identity line, indicating accurate recovery.

## A.5 Design Parameters and Practical Settings

This appendix provides practical guidance for selecting the design parameters of CMLR and outlines the specific settings used across all real-data experiments. A key advantage of CMLR is that it does not require extensive dataset-specific hyperparameter tuning. The main design parameters control numerical precision and computational efficiency rather than model capacity. These include:

1. the number of bins $T$ used in the Riemann approximation of the normalization constant,
2. the number of Fourier basis components $M$ used to represent the GP prior,
3. the mini-batch size $N'$ for stochastic optimization,
4. the learning rate $\alpha$ of the Adam optimizer, and
5. the decoding grid resolution $J$.

Table 2 summarizes the design parameter settings used in each real dataset, along with the number of neurons ($D$), number of trials ($N$), and output range ($\Omega$). These settings were fixed per dataset, but cross-validation can be incorporated into CMLR without modification. In practice, the two most important design parameters for numerical accuracy are the number of Riemann bins $T$ and the number of Fourier components $M$.

**Choice of $T$ (Riemann bins):** $T$ determines the numerical resolution used to approximate the normalization constant and should match the desired output-grid resolution. In practice, $T$ must be large enough that the density changes smoothly between adjacent grid points. For bounded or circular outputs, $T$ should scale with the width of the output range to avoid discretization artifacts. As a rule of thumb, for an output range of $[0, 1]$, choosing $T \approx 100$ provides adequate resolution, with larger values offering limited additional benefit.

**Choice of $M$ (Fourier components):** $M$ determines the spectral resolution of the Gaussian process prior. For smooth kernels such as the RBF, most of the spectral mass lies in low-frequency components, so only a small number of terms is required. Empirically, $M \in [15, 50]$ captures the relevant structure across all datasets while keeping computational cost low. As shown in Fig. S3, decoding performance is largely insensitive to $M$ beyond this range. Choosing a small $M$ is also desirable for reducing dimensionality and mitigating overfitting.

**Choice of $N'$ (mini-batch size):** The mini-batch size affects optimization speed rather than model capacity. Smaller batches increase the variance of stochastic gradients (which can aid exploration), whereas larger batches reduce noise but increase memory cost. We selected $N'$ to balance runtime and CPU memory constraints, with values between 1000 and 2000 working robustly.

**Choice of $\alpha$ (learning rate):** The learning rate determines the step size in stochastic variational optimization. Unlike $T$ and $M$, which control numerical resolution, $\alpha$ governs the stability and convergence speed of the optimizer. In practice, a moderately small value (e.g., $\alpha \in [10^{-3}, 10^{-1}]$) provides stable convergence across datasets. Larger values can lead to divergence, whereas smaller values slow training without improving accuracy. We selected $\alpha$ for each dataset using a small number of preliminary runs and found that its performance was largely insensitive within this range.

**Choice of $J$ (decoding grid resolution):** Finally, the number of decoding bins $J$ must be specified when evaluating predictive densities from the CMLR model, as is also the case for FlexCode and Naive Bayes. Since we observed a consistent trend of improved performance with finer decoding grids, we report all final results at the highest resolution tested, $J = 20,000$.

Table 2: Design parameter settings used in the real-data studies

| Dataset | $D$ (neurons) | $N$ (trials) | $\Omega$ (range) | $T$ | $M$ | $N'$ | $\alpha$ |
|---|---|---|---|---|---|---|---|
| Mouse V1 | 11311–20616 | 4282–4469 | $[0, 2\pi)$ | 100 | 17 | 1500 | 0.005 |
| Monkey V1 | 113–148 | 3600 | $[0, 2\pi)$ | 100 | 17 | 1500 | 0.005 |
| Mouse CA1 | 151–497 | 3600–5524 | $[0, 1]$ | 100 | 20 | 1400 | 0.02 |
| Monkey motor cortex | 164 | 25299 | $[-30, 30]^2$ | 1000 | 27 | 2000 | 0.1 |

## A.6 SCALABILITY OF CMLR WITH RESPECT TO NEURON COUNT AND DATASET SIZE

Here, we assess the robustness of CMLR on the mouse V1 dataset (data from Stringer et al. (2021)) under systematic variation of the neuron count ($D$) and dataset size ($N$), comparing its scaling behavior to XGBoost and DNN. Fig. S5 characterizes how decoding accuracy and training time scale under these manipulations. As the number of neurons increases (Fig. S5A), CMLR maintains stable absolute circular error and exhibits training times that grow approximately linearly with $D$, consistent with our simulation results in Appendix A.3. Across all values of $D$, CMLR achieves lower decoding error than XGBoost and DNN, and the performance gap widens at smaller $D$, indicating improved data efficiency. When the number of training samples is reduced (Fig. S5B), CMLR's performance degrades only slightly, whereas the data-driven baselines show more substantial deterioration. Training time for CMLR increases sublinearly with $N$ due to mini-batch stochastic variational inference. Overall, CMLR shows favorable computational and statistical scaling, and critically, it offers clear advantages over data-driven models in low-data regimes where principled structure and regularization matter most.

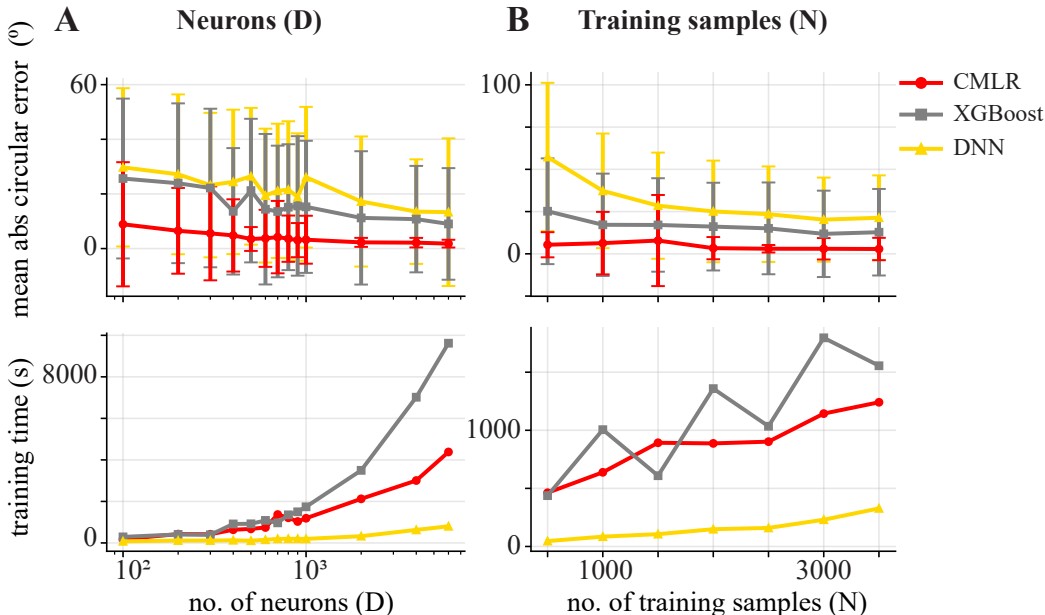

Figure S5: Scalability of decoding performance and training time for CMLR, XGBoost, and DNN on the mouse V1 dataset. Top: mean $\pm$ standard deviation of the absolute circular decoding error. Bottom: training time (seconds) measured on a standard laptop (Intel i7 2.4 GHz CPU, 16 GB RAM). Both metrics are shown as a function of (A) the number of neurons $D$ and (B) the number of training samples $N$.

### A.7 MACAQUE V1 DATA: DECODING DRIFTING GRATING ORIENTATIONS (EXTENSION OF SECTION 5.1)

Here, we show the results of applying our method to five electrophysiological recording datasets from the monkey primary visual cortex (data from Graf et al. (2011)), extending the results in Section 5.1. These datasets contained spiking activity recorded from between $D = 113$ and $D = 148$ neurons (treated as input features), and included 72 discrete stimulus orientations (outputs) spaced at $5°$ intervals, with 50 trials per orientation, resulting in a total of $N = 3600$ samples per dataset. Although our method is designed for continuous-valued stimuli, the discrete case is a special instance of the framework and is fully supported without modification. We applied the CMLR model using the same inference settings as in the mouse V1 dataset.

Fig. S6 summarizes the results. In panel A, CMLR and FlexCode consistently outperform Naive Bayes in mean absolute circular error across decoding resolutions, with all models plateauing beyond $J \approx 500$. Fig. S6B shows true versus decoded orientations for all models; CMLR predictions align closely with the identity line, and most large errors occur near $180°$, reflecting underlying bimodality. Panel C shows that FlexCode achieves the lowest decoding error (mean $\pm$ SD: $13 \pm 36°$, median: $3°$, IQR: $5°$), closely followed by CMLR ($14 \pm 42°$, $3°$, $4°$), with Naive Bayes performing moderately worse ($21 \pm 50°$, $4°$, $6°$), and XGBoost ($27 \pm 29°$, $17°$, $28°$), and DNN ($26 \pm 26°$, $20°$, $26°$) showing higher variability and error.

These results confirm the robustness of CMLR across both continuous and discretized decoding settings. Consistent with findings in the mouse V1 datasets, CMLR's correlation-aware formulation yields improved accuracy over correlation-blind baselines, reinforcing the importance of modeling shared variability in cortical populations for reliable neural decoding.

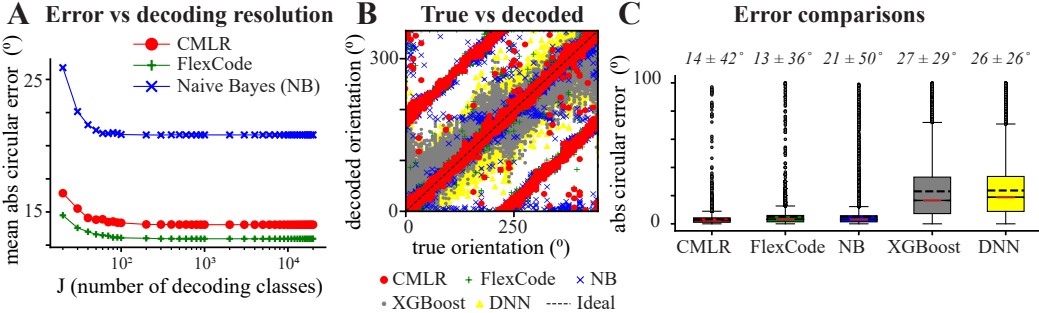

Figure S6: Application of the proposed method to macaque V1 data (data from Graf et al. (2011)). (A) Mean absolute circular error for CMLR, FlexCode, and Naive Bayes models as a function of the number of decoding classes $J$. (B) Scatter plots of decoded versus true stimulus orientations for CMLR, FlexCode, Naive Bayes, XGBoost, and DNN models. (C) Box plots comparing the absolute circular decoding error across all methods. Mean errors ($\pm$ standard deviation) are indicated as insets.

## A.8 Full decoding weight maps for monkey motor cortex data (extension of Fig. 4)

To provide a complete visualization of the model's output, we show the full set of 2D decoding weight functions inferred by CMLR for all 164 neurons in the monkey motor cortex dataset illustrated in Section 5.3. This Figure extends the subset shown in Fig. 4, revealing the rich diversity of 2D velocity tuning profiles captured by the model. The smooth, continuous weight surfaces illustrate how CMLR uncovers interpretable tuning structure across the two-dimensional velocity space, consistent with known motor cortex encoding properties.

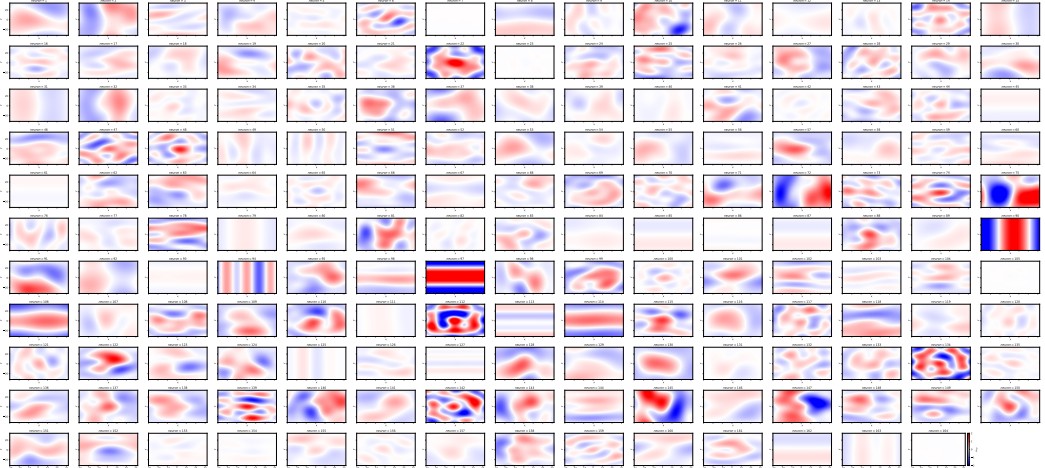

Figure S7: Full set of decoding weights inferred by CMLR for all $D = 164$ neurons in the monkey motor cortex dataset (data from Glaser et al. (2018; 2020)), extending the results in Fig. 4. Each subplot shows the neuron's two-dimensional tuning surface across the x and y velocity space. These results highlight the diversity and smoothness of velocity tuning captured by the model in real neural recordings.

## A.9 INTERPRETABILITY AND GENERALIZATION OF CMLR DECODING WEIGHT FUNCTIONS

In this Appendix, we further probe the CMLR decoding weight functions inferred for the motor-cortex velocity decoding task (data from Glaser et al. (2018; 2020)) shown in Fig. 4. Our goal is to evaluate both the interpretability of the inferred weights and their ability to generalize beyond the range of velocities observed during training. Fig. S8 presents two complementary analyses that address these questions by comparing the inferred weight functions to empirical neural tuning structure and by examining how the inferred functions behave when trained on progressively restricted subsets of the data.

First, we compared the inferred weight functions $w_d(y)$ with empirical spike–velocity associations derived directly from the training data (Fig. S8A). To obtain these empirical maps, we binned all training samples according to their two-dimensional velocity and, for each neuron $d$, accumulated that neuron's spike counts within each velocity bin. The resulting spike-weighted histograms were then smoothed and normalized to form empirical firing-rate density maps over the velocity space. These maps reveal the raw statistical structure of each neuron's tuning. They are strongly concentrated within the region of velocities that were actually visited during the experiment, which is typically a compact region around the origin, and they are often noisy and irregular because of finite sampling. In contrast, the CMLR inferred decoding weights exhibit smooth and spatially coherent tuning patterns defined over the entire velocity space. Importantly, in regions where empirical data are dense, the inferred weights align well with the empirical structure, capturing the dominant preferred directions and suppressive regions observed in the raw densities. This agreement indicates that the CMLR model faithfully extracts meaningful tuning characteristics from the data while simultaneously regularizing them into a smooth functional form that reflects the underlying relationship between velocity and firing activity.

Second, we studied how the inferred decoding weights behave when the training data are progressively restricted to smaller velocity ranges (Fig. S8B). By training the model using only samples within $|y^{(1)}|, |y^{(2)}| < 25, 20, 15, 10$, we artificially reduce the support of the observable data and thus increase the amount of extrapolation required outside the training domain. As the black boxes in Fig. S8B indicate, large portions of the velocity space are never observed when these restrictions are applied. Despite this, CMLR continues to produce well-structured and neuron-specific weight patterns that extend smoothly into unobserved regions. The fine details of the tuning curves become more dependent on the Gaussian process prior as the available data shrink, but the global structure remains consistent across all levels of data restriction. Preferred directionality, antagonistic regions, and smooth gradations remain stable as long as the model has at least partial coverage of the tuning landscape. This behavior reflects the inductive bias imposed by the smooth Gaussian process prior. When data are sparse or absent, the model defaults to the simplest and smoothest continuation that remains consistent with the observed samples.

Together, these analyses demonstrate two key properties of the CMLR framework:

1. **Interpretability**: the inferred decoding weights correspond closely to empirical neural firing statistics where data are available, offering intuitive and biologically meaningful descriptions of each neuron's response characteristics.

2. **Generalization**: the weight functions extend gracefully beyond the data-supported regions, providing coherent and stable extrapolations driven by the model's smoothness prior rather than fitting noise or artifacts.

These findings show that CMLR provides not only accurate decoding performance but also structured and interpretable functional maps that remain reliable when parts of the stimulus space are sparsely sampled. This is important for real-world neural decoding settings, because behavioral or sensory spaces are rarely sampled uniformly in practice.

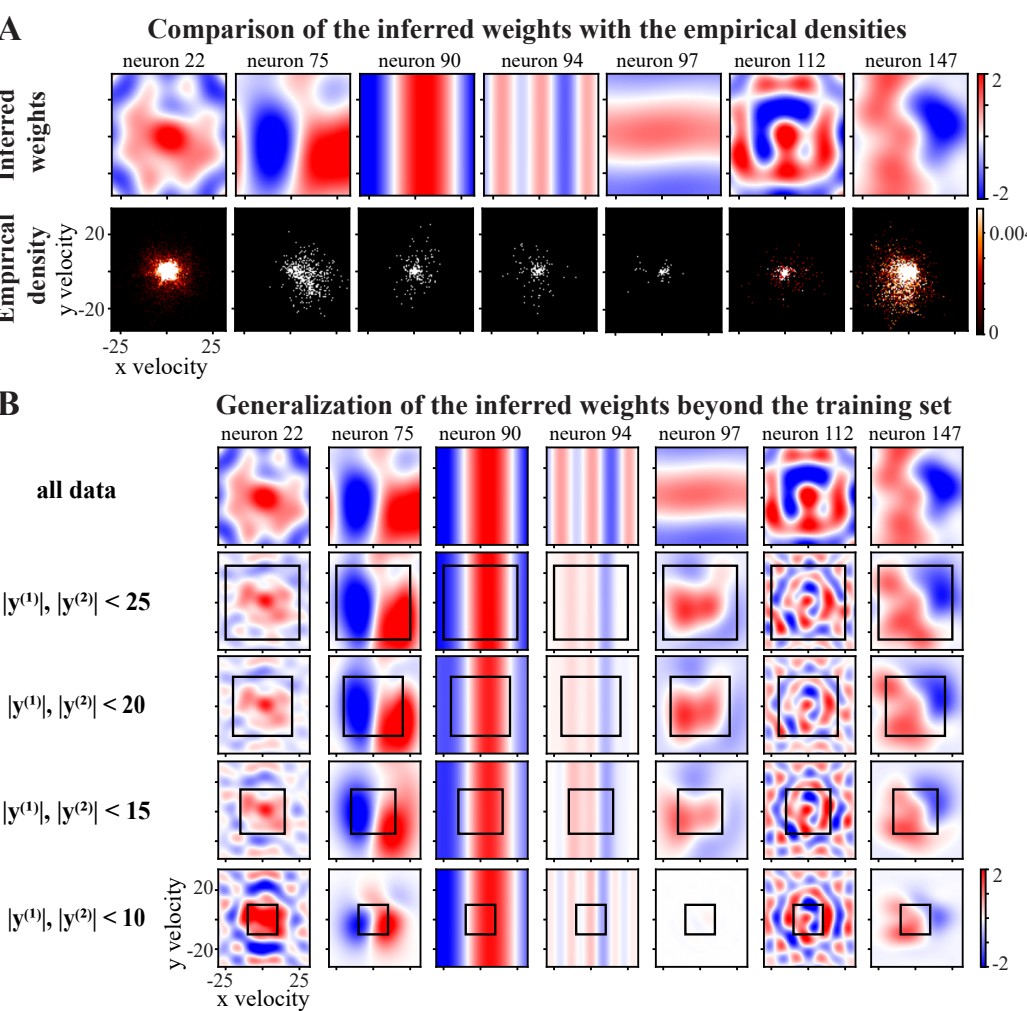

Figure S8: Interpretability and generalization of CMLR decoding weight functions for motor-cortex velocity decoding (data from Glaser et al. (2018; 2020)). (A) Inferred decoding weight functions for example neurons (top) compared with their empirical spike-weighted density maps (bottom). The empirical densities concentrate within the observed velocity range, while CMLR produces smooth, interpretable functions over the full space. (B) Generalization of inferred weights when training data are progressively restricted to smaller velocity bounds. Top to bottom: unconstrained, $|y^{(1)}|, |y^{(2)}| <$ 25, 20, 15, 10. Black boxes indicate the region of observed data. Even with limited training coverage, CMLR extrapolates smoothly outside the data-supported domain, demonstrating robustness and interpretability.

## A.10 CALIBRATION OF PREDICTIVE POSTERIORS

Here, we present the results of calibrating the posterior densities predicted by the CMLR model and FlexCode model on held-out test data. While the two conditional density estimation methods are trained to optimize the conditional likelihood, this does not necessarily guarantee that their predicted posteriors are well-calibrated; that is, that their uncertainty reflects empirical variability in the data. To assess calibration, we computed Probability Integral Transform (PIT) values (Dawid, 1984) and quantile calibration curves (Kuleshov et al., 2018), which diagnose whether predicted posterior distributions reflect the empirical distribution of true outputs. We also report the Expected Calibration Error (ECE) as a summary statistic.

### A.10.1 PROBABILITY INTEGRAL TRANSFORM (PIT)

The Probability Integral Transform (PIT) provides a diagnostic tool for assessing the calibration of continuous predictive distributions. For each test input $\mathbf{x}_n$, let $F_n(y)$ denote the predicted cumulative distribution function (CDF) for the output $y$. Given the true output $y_n$, the PIT value is defined as:

$$u_n = F_n(y_n)$$

which represents the cumulative probability mass assigned to values less than or equal to $y_n$ under the predicted posterior. If the model is perfectly calibrated, the PIT values $\{u_n\}_{n=1}^N$ should be independently and identically distributed as $\mathrm{Uniform}(0, 1)$.

Deviations from uniformity reveal miscalibration: U-shaped PIT histograms indicate overconfident predictions (posterior too narrow), hump-shaped histograms suggest underconfident predictions (posterior too wide), and asymmetric shapes reflect bias in the predictive distributions. Visual inspection of PIT histograms thus provides an interpretable diagnostic of calibration performance.

### A.10.2 QUANTILE CALIBRATION CURVES

Quantile calibration curves assess whether predicted quantiles contain the correct proportion of ground truth outputs. For each test input $\mathbf{x}_n$ and each nominal quantile level $\alpha \in (0, 1)$, we compute the $\alpha$-quantile $\hat{q}_\alpha^{(n)} = F_n^{-1}(\alpha)$ of the predicted posterior and then determine the empirical frequency with which the true output $y_n$ falls below this value. The empirical coverage at level $\alpha$ is defined as:

$$\mathrm{Coverage}(\alpha) = \frac{1}{N} \sum_{n=1}^N \mathbb{I}\left(y_n \leq \hat{q}_\alpha^{(n)}\right),$$

where $N$ is the number of test samples and $\mathbb{I}(\cdot)$ is the indicator function. Plotting empirical coverage versus nominal $\alpha$ yields the quantile calibration curve, with the diagonal representing perfect calibration.

To summarize calibration error across all quantile levels, we compute the expected calibration error (ECE) as:

$$\mathrm{ECE} = \frac{1}{|\mathcal{A}|} \sum_{\alpha \in \mathcal{A}} |\mathrm{Coverage}(\alpha) - \alpha|,$$

where $\mathcal{A}$ is a finite set of quantile levels (e.g., $\{0.05, 0.10, \ldots, 0.95\}$). A smaller ECE indicates better calibration.

### A.10.3 CALIBRATION ANALYSIS: CMLR VS FLEXCODE

We compared the uncertainty calibration of CMLR and FlexCode across three datasets (Mouse V1, Macaque V1, and Mouse CA1) using three diagnostics: PIT histograms, quantile calibration curves, and the relationship between PIT values and decoding error (Fig. S9). We restrict our calibration analysis to CMLR and FlexCode because calibration diagnostics such as PIT histograms and quantile calibration curves require access to the full conditional density. Regression-based models like XGBoost and DNN provide only point predictions rather than probability distributions, so PIT values, quantile calibration, and ECE cannot be computed for them.

**PIT histograms (Fig. S9A):** PIT histograms for CMLR are close to uniform across datasets, indicating that the model allocates probability mass in a way that matches the empirical distribution

of the observed outcomes. This reflects well-calibrated posterior predictions with no systematic concentration of mass in any region of the cumulative distribution. FlexCode, in contrast, shows clear deviations from uniformity, often exhibiting a central peak or a bimodal shape. These patterns indicate that FlexCode distributes probability mass unevenly and is therefore miscalibrated, assigning too much weight to certain regions of the output space relative to the true distribution.

**Quantile calibration curves (Fig. S9B):** CMLR's calibration curves closely follow the identity line across all datasets, with low ECE values (Mouse V1: $0.02 \pm 0.02$; Macaque V1: $0.03 \pm 0.01$; CA1: $0.04 \pm 0.01$), indicating accurate uncertainty quantification throughout the full quantile range. FlexCode shows larger deviations from the diagonal, with higher ECE values ($0.09 \pm 0.01$, $0.03 \pm 0.01$, $0.05 \pm 0.01$), consistent with its PIT histograms. Its empirical coverage typically falls below the ideal line at small quantiles and rises above it at large quantiles, indicating systematic miscalibration that produces under-coverage in the lower tail and over-coverage in the upper tail.

**PIT versus decoding error (Fig. S9C):** A well-calibrated model should show a clear relationship between PIT values and decoding error. In an ideal case, errors should be smallest near $\text{PIT} \approx 0.5$, where the model is most confident, and should increase smoothly as PIT approaches 0 or 1, where the model expresses greater uncertainty. This pattern indicates that the model's predictive distribution correctly reflects which points are easy or difficult to decode. Across all datasets, CMLR closely matches this ideal behavior: errors are lowest near $\text{PIT} \approx 0.5$ and rise gradually toward the extremes, demonstrating that the model's uncertainty estimates align with actual decoding difficulty. FlexCode, in contrast, shows elevated errors near $\text{PIT} \approx 0.5$, which is also where its PIT values are most concentrated, along with irregular fluctuations across the quantile range. This pattern indicates that FlexCode's predictive uncertainty does not reliably correspond to true error and that its diffuse predictions do not effectively communicate when mistakes are more likely.

**Summary:** Across all diagnostics and datasets, CMLR provides substantially better-calibrated and more informative uncertainty estimates than FlexCode. CMLR's posterior distributions align closely with empirical data and express uncertainty coherently, whereas FlexCode tends to produce overly diffuse densities that obscure predictive reliability. This demonstrates that CMLR not only improves decoding accuracy but also offers superior uncertainty quantification, which is essential for scientific and neural decoding applications.

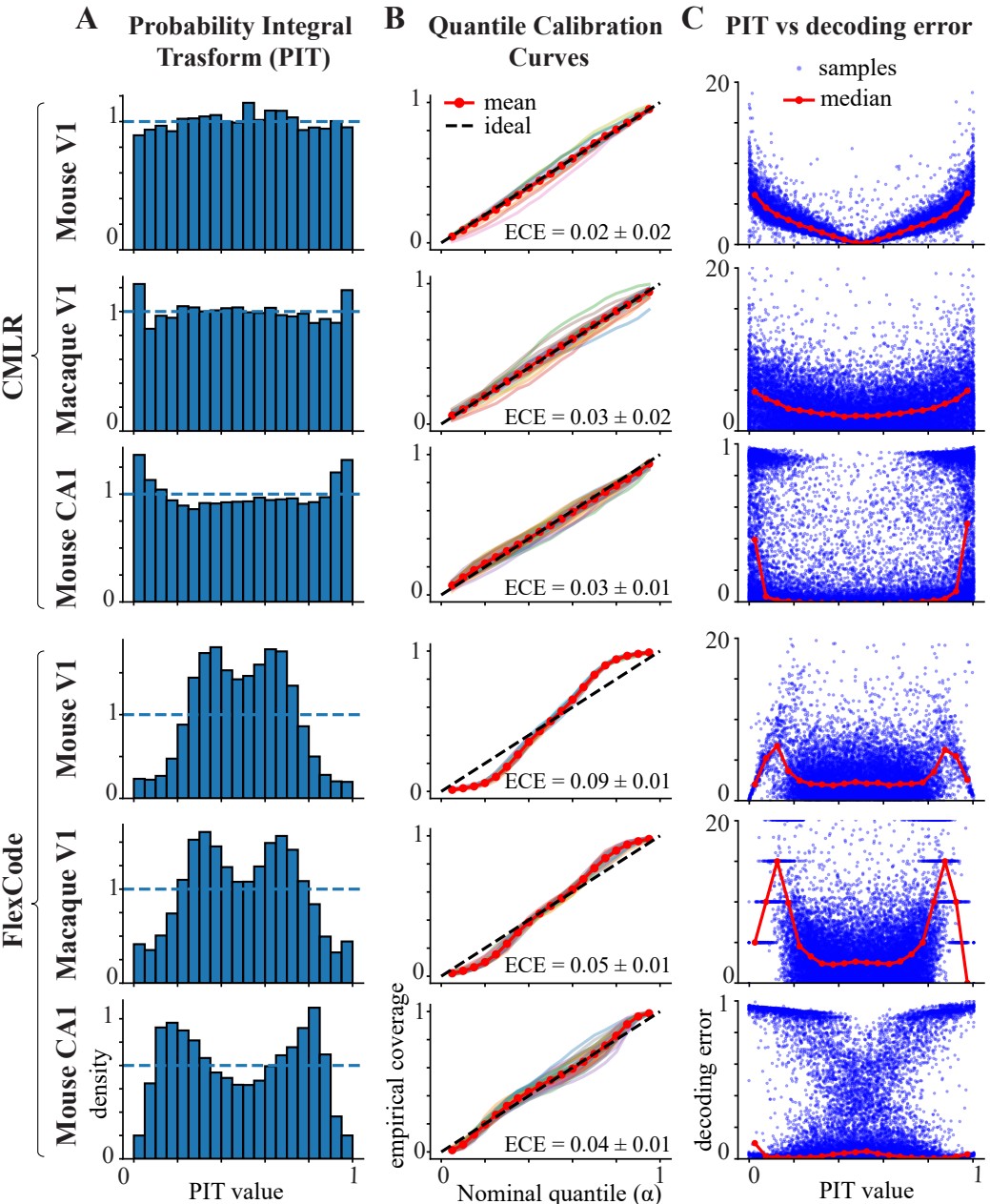

Figure S9: Calibration analysis for CMLR (top) and FlexCode (bottom) across Mouse V1, Macaque V1, and Mouse hippocampus CA1 datasets. (A) Probability integral transform (PIT) histograms. CMLR produces near-uniform histograms, whereas FlexCode exhibits clear non-uniform distributions. (B) Quantile calibration curves comparing empirical coverage to nominal quantile $\alpha$; the diagonal indicates perfect calibration. Insets report the expected calibration error (ECE; mean $\pm$ standard deviation across folds and datasets). (C) PIT value versus decoding error for individual test points, with median trends shown in red to illustrate how uncertainty relates to prediction error.

## A.11 Declaration of Usage of Large Language Models (LLMs)

We used large language models (LLMs) only for minor editing tasks, such as polishing grammar and improving readability. LLMs were not used to generate content, perform analyses, design methods, or conduct experiments. All scientific contributions and results in this manuscript are solely the work of the authors.

