# OpenReview forum: "Continuous multinomial logistic regression for neural decoding"
_ICLR.cc/2026/Conference — ICLR 2026 Poster_

### Official Review · Reviewer_Z69w · 2025-10-17

**Soundness:** 3
**Presentation:** 3
**Contribution:** 2
**Rating:** 6
**Confidence:** 3

**Summary:**

The paper introduces a combination of multinomial logistic regression with the gaussian processes defined on the weights.
The method is designed for neural decoding, where the goal is to estimate external variables from neural population activity, such as running speed or direction.
To handle computational complexity, the method assumed a univariate gaussian process per weight, uses standard radial basis function in the Fourier basis and trains the model as a stochastic variational inference.
Then they test the model on several neural datasets, comparing with XGBoost and DNNs.

**Strengths:**

1. **Clarity**. The paper is clearly written and suggests extensive (theoretical) comparison with prior work (*"Connections to Prior Work"* section)
2. **Various datasets**. The papers tests their method on various datasets from different species and data modalities (calcium imaging and electrophysiology)
3. **Provided implementation**. The code is attached in the supplementary material.
4. **Strong baselines**. The baselines are chosen to cover both bayesian and performance-drive methods (like DNN), covering the field.

**Weaknesses:**

1. **Unclear computational scalability**. The paper claims to provide a *"scalable framework"*, however, the practical aspects of scalability are under-explored. What are the computational restrictions? How many additional parameters the method provides and how longer does it take to train compared to modelling regression with uncertainty or the other baselines provided (such as FlexCode or XGBoost)?
2. **Baselines**.The comparison with standard multinomial logistic regression is missing.  For the other strong baselines, it is not clear how they were optimized per dataset if any optimisations were present.
3. **Lack of ablations for design choices justifications**.  While authors acknowledge in limitations that multivariate Gaussian processes could be used and fixed Fourier-domain bases and RBF kernels could be replaces by adaptive basis functions - all of these are not conceptual limitations of the methods but rather a list of sometimes very straightforward technical improvements (like multivariate Gaussian processes) and could be done within the current submission.
4. **Unclear interpretability gains**. See Q6 -  What are the additional interpretability benefits provided by CMLR? XGBoost can also give an interpretable weighted impact of each neuron on the target variable. Multinomial logistic regression with uncertainty can also be close in terms of interpretations.

**Questions:**

1. How is CMLR related to the following works [1-3]?
2. Why the XGBoost and DNNs lines are missing in Fig 2c and Fig 3A?
3. I might have missed it in the text but which loss function do you use to train the CMLR? Is it MSE loss everywhere?
4. Why there is no comparison with the standard multinomial logistic regression? You do not really analyse uncertainties in the main text, and even for uncertainties methods like Laplace Redux [3] could be used to derive uncertainties for a non-bayesian methods.
5. Have you tuned the hyperparameters of DNN and XGBoost per dataset? As your datasets had different sizes the ratio of data to parameters might be crucial for performance. How DNN parameters compared to the CMLR learn parameters?
6. What are the additional interpretability benefits provided by CMLR? XGBoost can also give an interpretable weighted impact of each neuron on the target variable.

References:
[1] Chan, Antoni B. "Multivariate generalized gaussian process models." arXiv preprint arXiv:1311.0360 (2013).
[2] Payne, Richard D., et al. "A conditional density estimation partition model using logistic Gaussian processes." Biometrika 107.1 (2020): 173-190.
[3] Daxberger, Erik, et al. "Laplace redux-effortless bayesian deep learning." Advances in neural information processing systems 34 (2021): 20089-20103.
[3] Murray, Iain, David MacKay, and Ryan P. Adams. "The Gaussian process density sampler." Advances in neural information processing systems 21 (2008).

---

> ### Author Response · Authors · 2025-11-20
>
> We sincerely thank the reviewer for the thorough and constructive feedback. We appreciate the recognition of the clarity of the manuscript, the breadth of datasets, the strong baselines, and the provided implementation. Below, we address all concerns and questions point-by-point and summarize the revisions made to the manuscript.
>
> # Computational scalability and practical restrictions
>
> We agree that the original manuscript did not sufficiently articulate the computational considerations underlying CMLR. We thank the reviewer for raising this important point. In the revised manuscript (**Section 3.3**, p. 4, lines 204–211), we now explicitly describe how the computational complexity of CMLR scales with the key parameters, based on the results in Appendix A.3 (Fig. S3). In particular:
> - Training complexity scales approximately *linearly* in $D$ (number of neurons/features).
> - Training complexity scales *sublinearly* in $N$ (number of samples), due to the use of mini-batch SVI.
>
> Furthermore, unlike data-driven methods, CMLR does not require extensive dataset-specific hyperparameter tuning. We have added a new section (**Appendix A.5**) that provides practical guidance for selecting all CMLR design parameters and clarifies that these settings influence numerical precision and computational efficiency rather than model capacity. Specifically, the appendix now explains:
> 1. **Riemann bins $T$:** how $T$ determines the resolution of the normalization integral, with principled guidelines for selecting $T$ based on output-range width (e.g., $T \approx 100$ for outputs in $[0,1]$).
> 2. **Fourier components $M$:** how $M$ controls the spectral resolution of the GP prior, why only relatively low-frequency components are needed for our datasets (i.e., due to the empirical range of neural tuning bandwidths), and why values in the range $M \in[15,50]$ provide stable performance while keeping computation low.
> 3. **Mini-batch size $N’$:** how it affects optimization speed rather than model capacity, with practical ranges $(1000–2000)$ that balance runtime and memory.
> 4. **Learning rate $\alpha$:** how $\alpha$ governs optimizer stability, with guidance that moderately small values $\alpha \in [10^{-3}, 10^{-1}]$ provide reliable convergence.
> 5. **Decoding grid resolution $J$:** how $J$ determines the evaluation grid for predictive densities, with final results reported at the finest resolution tested ($J=20,000$).
>
> Appendix A.5 also includes **Table 2**, which reports the specific parameter settings used for each real-data experiment. Finally, the Discussion now notes that nonlinear baselines typically require extensive hyperparameter tuning, whereas CMLR’s design parameters require minimal dataset-specific adjustment, which is a practical advantage of the framework.
>
> To complement the complexity discussion, we have added a runtime comparison across all methods and datasets (**Section 5.4; Table 1**), which shows that CMLR remains computationally practical across diverse settings. In addition, a new scalability analysis in **Appendix A.6** (**Fig. S5**) evaluates how runtime and accuracy scale with $D$ and $N$ on Mouse V1 data, demonstrating favorable scaling behavior.
>
> # Design choices
>
> We appreciate the reviewer’s thoughtful comments. Our goal in this work is to introduce a principled continuous extension of MLR that balances interpretability, scalability, and analytical tractability. Accordingly:
> - **Independent GP priors** preserve neuron-specific tuning curves, a core interpretability feature that would be obscured under multivariate GP priors due to induced coupling across neurons.
> - The **RBF kernel** can be diagonalized in the Fourier domain and makes SVI scalable to tens of thousands of neurons by avoiding kernel inversions and operating in a reduced-dimensional space. Alternative kernels or adaptive basis functions would require nontrivial redesign of the inference scheme and would likely compromise computational efficiency.
>
> We now emphasize in the revised manuscript (**Discussion**, p. 10, lines 520-526) that these extensions are promising future directions, but adopting them in the current framework would come at the expense of interpretability or scalability.

---

> > ### Author Response · Authors · 2025-11-20
> >
> > # Responses to specific questions
> >
> > ## Q1. Relationship to works [1–4]:
> >
> > We have expanded the **Connections to Prior Work** (p. 10) section to explicitly compare CMLR with the cited references:
> > 1. **Chan (2013):** develops multivariate output regression GP models but does not address conditional density estimation. In contrast, CMLR is a CDE method that facilitates direct assessment of uncertainty, calibration, and the structure of the conditional density.
> > 2. **Payne et al. (2020):** propose a partition-based CDE model using logistic Gaussian processes, but the approach does not include the additive structure or the scalable variational inference framework used in CMLR. As a result, it does not scale to datasets with thousands of neurons and does not provide the neuron-specific tuning functions that make CMLR interpretable.
> > 3. **Daxberger et al. (2021):** develop Laplace approximations for Bayesian neural networks, but these methods provide approximate uncertainty around point predictions rather than full conditional densities over continuous variables. Such point-estimator uncertainty cannot capture the rich structure that conditional densities provide, including skewness, multimodality, and circular or bounded support.
> > 4. **Murray et al. (2008):** introduce the Gaussian Process Density Sampler, which performs unconditional density estimation (not CDE) using MCMC over latent GP values. This approach targets a different problem from CMLR and does not scale to high-dimensional conditional settings such as neural decoding, nor does it provide the additive structure or interpretability that CMLR offers.
> >
> > ## Q2. Missing XGBoost and DNN in Figs. 2C and 3A:
> >
> > As clarified in the revised manuscript (**Section 5.1**, p. 6, lines 307–309):
> > - These panels evaluate how decoding error changes with decoding resolution $J$.
> > - Only **conditional density estimators** (CMLR, FlexCode, Naive Bayes) define a full conditional density and can therefore be evaluated at arbitrary $J$ by construction (see Section 4).
> > - In contrast, XGBoost and DNN produce **point predictions** rather than conditional densities, so there is no meaningful way to vary $J$ or include them in resolution-dependent analyses.
> >
> > ## Q3. Loss function:
> >
> > CMLR is trained by maximizing the **Evidence Lower Bound** (ELBO) (Eq. 1, Section 3), which is the standard variational inference objective. ELBO is therefore the training objective, not MSE. This is now explicitly stated (**Section 3**, p. 3, line 154).
> >
> > ## Q4. Comparison with standard multinomial logistic regression (MLR):
> >
> > We appreciate the suggestion to include this baseline. The most directly comparable model is discrete MLR with Gaussian process priors over the decoding weights, as studied in Greenidge et al. (2024), which showed that GP-regularized MLR substantially outperforms classical unregularized MLR (as well as MLR with Elastic-Net penalties) on discretized neural decoding tasks. CMLR can be viewed as a continuous generalization of this framework: if the output domain is discretized into a small number of bins (small $J$), CMLR reduces to GP-regularized MLR in Greenidge et al. (2024). As shown in Fig. 3C, decoding error decreases as the bin resolution increases, illustrating how CMLR extends MLR into the continuous limit and achieves superior performance. Unlike standard MLR, CMLR supports arbitrarily fine output resolutions and enables inference directly in continuous space, providing greater flexibility for modeling circular, bounded, and high-resolution continuous variables. We now make this relationship explicit in the revised manuscript (**Section 5.1**, p. 6, lines 323-357).

---

> > > ### Author Response · Authors · 2025-11-20
> > >
> > > ## Q5. Hyperparameter tuning of baselines:
> > >
> > > We fully agree that transparent and fair comparisons are essential. The revised manuscript now includes:
> > > - Full **5-fold cross-validation** for all datasets and methods.
> > > - **Bayesian optimization** [Gardner et al., 2014] of hyperparameters within each fold , using a 64–16–20 train/validation/test split (as in Glaser et al., 2020).
> > > - **Hyperparameter search spaces**:
> > > 1. XGBoost: tree depth, number of trees, learning rate
> > > 2. DNN: number of hidden units, dropout rate, number of epochs
> > >
> > > The procedure is now clearly described in the revised manuscript (pp. 5–6), and the updated **Figs. 2, 3, 4, and S6** reflect the tuned results. CMLR continues to outperform XGBoost and DNN on Mouse V1 (Fig. 2), Monkey V1 (Fig. S6), and Mouse CA1 (Fig. 3), though with somewhat reduced performance gaps. These datasets all fall in regimes where *sample sizes are moderate relative to dimensionality* and where the output variable exhibits a *bounded structure*. In such settings, CMLR benefits from its **GP-based functional priors and additive structure**, which provide strong regularization, promote smoothness in the decoded variable, and mitigate overfitting when data are limited. In contrast, XGBoost and DNNs require larger sample sizes to fully leverage their capacity and tend to be more sensitive to noise and trial-to-trial variability in these datasets.
> > >
> > > Furthermore, the comparatively larger performance gap observed for circular orientation decoding (Mouse V1 and Monkey V1) can be attributed to several additional factors:
> > > - **CMLR explicitly models the bounded and circular nature of orientation**, whereas XGBoost and DNN do not incorporate circular geometry.
> > > - **Conditional density estimation (CDE) provides full predictive distributions**, allowing CMLR to represent periodic and multimodal structure (and we observed underlying bimodality in these datasets). Point-estimate models cannot capture these features.
> > > - **XGBoost and DNN performance depends critically on incorporating activity from adjacent time bins** (as in Mouse CA1 and Monkey MC). In the V1 datasets, however, adjacent orientations are not continuous due to the design of the experiment, limiting the benefit of temporal or contextual information for these models.
> > >
> > > We now explicitly discuss these points in the revised manuscript (**Introduction**, p. 2, lines 86–92).
> > >
> > > For Monkey MC (Fig. 4), XGBoost and DNN now slightly outperform CMLR. We feel this is perhaps unsurprising given the very large training set that favors high-capacity nonlinear models. (We thank the reviewer for the suggestion to carefully optimize the hyperparameters for these two methods, which has indeed changed the relative ordering of methods for this dataset!)  To further analyze dataset-size effects, we have added a new scalability study on the Mouse V1 dataset (**Appendix A.6**, Fig. S5), which shows that CMLR experiences only modest reductions in accuracy as the number of neurons $D$ and samples $N$ decrease while continuing to scale efficiently. In these lower-data regimes, the performance of CMLR improves substantially relative to XGBoost and DNN. These updated results provide a clearer and fairer comparison across methods and illustrate the settings in which CMLR offers the greatest advantage. We thank the reviewer for highlighting the importance of these adjustments and believe they significantly strengthen the robustness of our findings.
> > >
> > > ## Q6: Interpretability benefits of CMLR
> > >
> > > We thank the reviewer for raising this point. We clarify the interpretability advantages in the revised manuscript. Although XGBoost can provide global feature importance scores, these values summarize contributions across the entire output space and do not reveal how a neuron’s influence changes as a function of the continuous variable being decoded. In contrast, CMLR produces an explicit decoding weight function $w_d(y)$ for each neuron, which serves as a **continuous tuning curve over the output domain**. These curves can be directly visualized, compared across neurons, and interpreted in the same way as **classical systems-neuroscience tuning curves**.
> > >
> > > Furthermore, because CMLR is a conditional density estimation method, it yields **full predictive distributions** rather than point estimates. This enables direct assessment of uncertainty, calibration, and the structure of the conditional density, including circular, multimodal, or asymmetric characteristics that point-estimate models such as XGBoost or DNNs cannot capture, even with post hoc uncertainty approximations. We highlight these interpretability and uncertainty advantages more clearly in the revised manuscript (**Discussion**, p. 9, lines 476–485).

---

> > > > ### Author Response · Authors · 2025-11-20
> > > >
> > > > # Concluding remarks
> > > >
> > > > We thank the reviewer once again for the insightful and constructive feedback. The revised manuscript now incorporates substantial improvements, including:
> > > > - clearer computational analysis,
> > > > - strengthened baselines through comprehensive hyperparameter tuning,
> > > > - explicit comparisons to related work,
> > > > - extended scalability and uncertainty analyses, and
> > > > - clarification of interpretability advantages.
> > > >
> > > > We believe these additions significantly enhance the rigor, clarity, and scientific value of the work.
> > > >
> > > > ## References
> > > >
> > > > Gardner, J., Kusner, M., Xu, Z., Weinberger, K., & Cunningham, J. (2014). Bayesian optimization with inequality constraints. ICML, 937–945.
> > > >
> > > > Glaser, J. et al. (2020). Machine learning for neural decoding. eNeuro, 7(4).
> > > >
> > > > Greenidge, C. D., Scholl, B., Yates, J. L., & Pillow, J. W. (2024). Efficient decoding of large-scale neural population responses with Gaussian-process multiclass regression. Neural Computation, 36(2), 175–226.

---

> > > > > ### Comment · Reviewer_Z69w · 2025-11-25
> > > > > **Interepretability follow-up**
> > > > >
> > > > > Thanks a lot for clear answers and constructively addressing the criticism especially wrt to baselines and runtime.
> > > > >
> > > > > I have a follow-up question about interpretability. I do get that CMLR produces an explicit decoding weight function $w_d(y)$, however, I am missing the validation that these functions are meaningful, and generalize well outside the range of $y$ seen during training. A suggestion - for fig 4A you plot the inferred density maps per neuron - is it possible to also plot the empiric density maps below. If I understand it correctly, this should address my concert to validate that the functions are meaningful - or do I get something wrong?
> > > > > As for the generalization, this might require a small additional experiment to remove all values of x and y above a certain thresholds from the train set and compare the infered maps again, highlighting the removed experiment. (quantifying the difference between empiric and infered density map could be an additional follow up if the visual analysis is unclear.)

---

> > > > > > ### Author Response · Authors · 2025-11-27
> > > > > >
> > > > > > Thank you very much for the constructive comments and for raising important points regarding interpretability and generalization. We fully agree that validating the meaning of the inferred decoding weight functions is essential. In the revised manuscript, we incorporated the exact analyses the reviewer suggested. We added a new dedicated section (Appendix A.9) with a detailed discussion, accompanied by the new Figure S8, and we briefly reference these results in the main text (Section 5.3: p. 8, lines 405–407).
> > > > > >
> > > > > > **Empirical density maps**: To assess whether the inferred decoding weights reflect the empirical structure of the data, we computed empirical spike–velocity density maps for each neuron by binning spikes according to the two-dimensional hand velocity and smoothing and normalizing the resulting histograms. We then plotted these empirical maps directly below the inferred CMLR decoding weights (Fig. S8A). The empirical maps are strongly concentrated within the region of velocities that were actually visited during the experiment and are often noisy because of finite sampling. In contrast, the CMLR inferred weights exhibit smooth and spatially coherent patterns over the entire velocity space. Within regions supported by training data, the inferred weights align well with the empirical maps, capturing the dominant preferred directions and suppressive regions. This demonstrates that CMLR extracts meaningful structure from the data while regularizing it into a smooth functional representation of the relationship between velocity and firing activity.
> > > > > >
> > > > > > **Generalization outside the observed range**: We also carried out the reviewer’s proposed experiment by systematically removing all samples with $|y^{(1)}|$ and $|y^{(2)}|$ above several thresholds (25, 20, 15, and 10) and re-estimating the decoding weights using only the restricted training sets. The resulting maps are shown in Fig. S8B. Although large portions of the velocity space are withheld during training, the inferred weight functions remain smooth and coherent outside the observed region. Within the restricted region, they match the empirical structure, and outside it, they extrapolate in a manner consistent with the smoothness prior of the Gaussian process. As the available data shrink, the fine details naturally depend more on the prior, but the global structure remains stable across all levels of restriction. Preferred directionality, antagonistic regions, and gradual transitions remain consistent, demonstrating that CMLR generalizes in a stable and interpretable way, even when substantial parts of the output space are unobserved.
> > > > > >
> > > > > > **Clarification on the role of the weight functions**: We also wish to clarify that the CMLR weight functions are not neural tuning curves in the literal sense of a neural encoding model. They are conceptually analogous to neural tuning curves because they show how each neuron's activity shapes the predicted density over the variable of interest, although they arise from a decoding model and do not reflect the neuron's encoding properties. Naive Bayes, which fits each neuron independently, is more directly aligned with traditional tuning curves. A central motivation for CMLR is that decoding based solely on independent tuning curves, as in Naive Bayes, performs poorly because it ignores correlations between neurons, whereas CMLR captures joint structure across neurons. We clarified this point in the revised manuscript (Introduction, p. 1, lines 51–53).
> > > > > >
> > > > > > **In summary, we addressed the reviewer’s comment by**:
> > > > > > 1. Comparing the inferred decoding weights with empirical spike–velocity maps.
> > > > > > 2. Testing generalization by removing large regions of the training set and re-estimating the weights.
> > > > > > 3. Clarifying the conceptual relation between CMLR weight functions and tuning curves.
> > > > > >
> > > > > > We appreciate the reviewer’s suggestion, and believe that the new analyses directly address the concern. These results support that CMLR produces decoding weight functions that are meaningful, robust, and consistent with empirical structure. We sincerely thank the reviewer for this insightful request.

---

### Official Review · Reviewer_4VSC · 2025-10-28

**Soundness:** 3
**Presentation:** 4
**Contribution:** 4
**Rating:** 6
**Confidence:** 4

**Summary:**

The authors introduce continuous multinomial logistic regression (CMLR), a flexible nonparametric model that allows for both discrete- and continuous-valued outputs by mapping inputs to a full probability density using per-feature additive functions with Gaussian priors. The resulting model is applied to a wide variety of neural decoding tasks, both continuous and discrete, and shows impressive results compared to baselines.

**Strengths:**

The paper is well-written and clearly articulates both the gap in the current literature and the exposition of the method. As applied to the neural decoding problem, the weight functions become per-neuron tuning curves that provide more interpretability than tree-based or neural network approaches. The method is applied to a range of neural decoding problems (sensory and motor) and shows strong performance across all datasets. The limitations and future directions section makes clear that this is a rich model class with many exciting directions to explore, and the paper is a strong foundation from which to start this work.

**Weaknesses:**

My main concern is the thoroughness of the baseline comparisons. The authors chose a different number of Fourier components M and mini-batch size N' for each of their datasets, but did not clearly state how these values were chosen. Furthermore, it seems no attempt was made to search hyperparameter space for XGBoost or the DNN. Better hyperparameter tuning (especially for the baselines) would therefore give me more confidence in the stated results. The authors could, for example, choose 2 or 3 important hyperparameters per method and search over these by performing a train/val split of the 80% of training data and selecting the best hyperparameters on the validation data.

I am also concerned about the robustness of the results if they are indeed only reported on 20% of the data. My own experience in neural decoding has been that the train/test split can significantly affect model performance, and conclusions from one split may not hold with another split. Performing full k-fold cross validation (where every trial lands in the test set exactly once) would better control for noise introduced in the sampling process.

**Questions:**

What is the computational complexity of CMLR, i.e. how do training and inference time scale with D, T, M, J? The authors have included training times in the Appendix, but it might be useful to mention these numbers (or at least their order of magnitude) more explicitly in the main text.

How much data do I actually need to train CMLR? Fig S3 is very cool and helpful, and it would be interesting to see something similar with real data. Do the GP priors/additive structure of CMLR make it more amenable to fitting models with less data, as compared to XGBoost or DNNs?

Fig 2E/3C/4C/S4C: what is the value of J used for CMLR/FlexCode/NB? How was this value chosen?

I find the brief mention of uncertainty calibration very interesting; it is well-known that DNNs, for example, are very often poorly calibrated. It would be cool to see the PIT results (i.e. Fig S5) for some of the other methods. Is it possible to do this for DNNs? Better calibration could be as strong a selling point as better accuracy for scientific applications, and might be worth emphasizing this in the main text more.

More with uncertainty calibration: having never seen PIT histograms, I can believe they are a good approach for quantifying calibration, but they feel very disconnected from the actual datasets that are being analyzed. The mouse hippocampal data offers an interesting example: here CMLR tends to make mistakes when the true or decoded position is at one end of the track. What do uncertainty estimates look like for these mistakes? Does the uncertainty scale with the magnitude of the error? How do those compare to uncertainty estimates from the other methods?

Minor: not having a background in this literature, the following sentences in the first Intro paragraph were a bit confusing to me: "However, many neural decoding tasks involve continuous variables... . In such settings, researchers who wish to use MLR-like decoding methods are commonly forced to discretize the output variable into a finite number of classes." At this point I don't really know what "MLR-like decoding methods" are and the first thing I think of is "sure but doesn't linear regression work just fine?" Perhaps clarifying this point early will help convince readers unfamiliar with the literature.

---

> ### Author Response · Authors · 2025-11-20
>
> We sincerely thank the reviewer for the thoughtful and constructive feedback. We appreciate the recognition of the clarity, novelty, and breadth of our contributions, as well as the emphasis on CMLR’s interpretability and potential as a foundation for future work. Below, we address the reviewer’s concerns and questions point-by-point and summarize the revisions made.
>
> # Hyperparameter tuning and robustness of baseline comparisons
>
> We appreciate the reviewer raising this important point and fully agree that thorough and fair baseline comparisons are essential. The original manuscript did not clearly describe this procedure, and we have addressed this in the revised version. Specifically, we have now:
> - Performed **full 5-fold cross-validation** for all datasets, ensuring that each trial appears exactly once in the test set.
> - Clarified the selection of **CMLR design parameters** ($T$, $M$, $N'$) in Appendix A.5 (p. 22) and summarized them in Table 2, noting that these settings were fixed per dataset.
> - Optimized XGBoost and DNN hyperparameters via **Bayesian optimization** [Gardner et al., 2014] within each fold, using a 64–16–20 train/validation/test split, following best practices from Glaser et al. (2020).
> - **Searched over key hyperparameters**:
> 1. XGBoost: tree depth, number of trees, learning rate
> 2. DNN: number of hidden units, dropout rate, number of epochs
>
> The procedure is now clearly described in the revised manuscript (pp. 5–6), and the updated **Figs. 2, 3, 4, and S6** reflect the tuned results. CMLR continues to outperform XGBoost and DNN on Mouse V1 (Fig. 2), Monkey V1 (Fig. S6), and Mouse CA1 (Fig. 3), though with somewhat reduced performance gaps. These datasets all fall in regimes where *sample sizes are moderate relative to dimensionality* and where the output variable exhibits a *bounded structure*. In such settings, CMLR benefits from its **GP-based functional priors and additive structure**, which provide strong regularization, promote smoothness in the decoded variable, and mitigate overfitting when data are limited. In contrast, XGBoost and DNNs require larger sample sizes to fully leverage their capacity and tend to be more sensitive to noise and trial-to-trial variability in these datasets.
>
> Furthermore, the comparatively larger performance gap observed for circular orientation decoding (Mouse V1 and Monkey V1) can be attributed to several additional factors:
> - **CMLR explicitly models the bounded and circular nature of orientation**, whereas XGBoost and DNN do not incorporate circular geometry.
> - **Conditional density estimation (CDE) provides full predictive distributions**, allowing CMLR to represent periodic and multimodal structure (and we observed underlying bimodality in these datasets). Point-estimate models cannot capture these features.
> - **XGBoost and DNN performance depends critically on incorporating activity from adjacent time bins** (as in Mouse CA1 and Monkey MC). In the V1 datasets, however, adjacent orientations are not continuous due to the design of the experiment, limiting the benefit of temporal or contextual information for these models.
>
> We now explicitly discuss these points in the revised manuscript (**Introduction**, p. 2, lines 86–92).
>
> For Monkey MC (Fig. 4), XGBoost and DNN now slightly outperform CMLR. We feel this is perhaps unsurprising given the very large training set that favors high-capacity nonlinear models. (We thank the reviewer for the suggestion to carefully optimize the hyperparameters for these two methods, which has indeed changed the relative ordering of methods for this dataset!)  To further analyze dataset-size effects, we have added a new scalability study on the Mouse V1 dataset (**Appendix A.6**, Fig. S5), which shows that CMLR experiences only modest reductions in accuracy as the number of neurons $D$ and samples $N$ decrease while continuing to scale efficiently. In these lower-data regimes, the performance of CMLR improves substantially relative to XGBoost and DNN. These updated results provide a clearer and fairer comparison across methods and illustrate the settings in which CMLR offers the greatest advantage. We thank the reviewer for highlighting the importance of these adjustments and believe they significantly strengthen the robustness of our findings.

---

> > ### Author Response · Authors · 2025-11-20
> >
> > # Responses to specific questions
> >
> > ## Q1. Computational complexity and runtime
> >
> > We thank the reviewer for raising this important point. In the revised manuscript (**Section 3.3**, p. 4, lines 204–211), we now explicitly describe how the computational complexity of CMLR scales with the key parameters, based on the results in Appendix A.3 (Fig. S3). In particular:
> > - Training complexity scales approximately *linearly* in $D$ (number of neurons/features).
> > - Training complexity scales *sublinearly* in $N$ (number of samples), due to the use of mini-batch SVI.
> >
> > Furthermore,unlike data-driven methods, CMLR does not require extensive dataset-specific hyperparameter tuning. We have added a new section (**Appendix A.5**) that provides practical guidance for selecting all CMLR design parameters and clarifies that these settings influence numerical precision and computational efficiency rather than model capacity. Specifically, the appendix now explains:
> >
> > 1. **Riemann bins $T$:** how $T$ determines the resolution of the normalization integral, with principled guidelines for selecting $T$ based on output-range width (e.g., $T \approx 100$ for outputs in $[0,1]$).
> > 2. **Fourier components $M$:** how $M$ controls the spectral resolution of the GP prior, why only low-frequency components are needed for RBF kernels, and why values in the range $M \in[15,50] $ provide stable performance while keeping computation low.
> > 3. **Mini-batch size $N’$:** how it affects optimization speed rather than model capacity, with practical ranges $(1000–2000)$ that balance runtime and memory.
> > 4. **Learning rate $\alpha$:** how $\alpha$ governs optimizer stability, with guidance that moderately small values $\alpha \in [10^{-3}, 10^{-1}]$ provide reliable convergence.
> > 5. **Decoding grid resolution $J$:** how $J$ determines the evaluation grid for predictive densities, with final results reported at the finest resolution tested ($J=20,000$).
> >
> > Appendix A.5 also includes **Table 2**, which reports the specific parameter settings used for each real-data experiment. Finally, the Discussion now notes that nonlinear baselines typically require extensive hyperparameter tuning, whereas CMLR’s design parameters require minimal dataset-specific adjustment, which is a practical advantage of the framework.
> >
> > To complement the complexity discussion, we added a runtime comparison across all methods and datasets (**Section 5.4**; **Table 1**), which shows that CMLR remains computationally practical across diverse settings. In addition, a new scalability analysis in **Appendix A.6** (Fig. S5) evaluates how runtime and accuracy scale with $D$ and $N$ on Mouse V1 data, demonstrating favorable scaling behavior.
> >
> > ## Q2. Data efficiency and real-data learning curves:
> >
> > We fully agree that data efficiency is an important consideration and appreciate the reviewer’s suggestion. In response, we added a real-data scalability analysis for the mouse V1 dataset (new **Appendix A.6**;  Fig. S5), evaluating CMLR, XGBoost, and DNN performance as a function of both the number of neurons ($D$) and the number of training samples ($N$).
> >
> > The new results show that CMLR maintains stable decoding accuracy even when the available data are substantially reduced, and it consistently outperforms the baselines in low-data regimes. Moreover, the performance gap between CMLR and the data-driven baselines widens as $D$ and $N$ decrease, consistent with the reviewer’s intuition regarding the advantages conferred by GP priors and the additive structure of the model.
> >
> > ## Q3. Choice of decoding resolution  $J$:
> >
> > We now explicitly state in **Appendix A.5** that all final results are reported using the finest decoding grid tested, $J = 20,000$, which gives stable and high-resolution density estimates for CMLR, FlexCode, and NB.
> >
> > ## Q4. Uncertainty calibration and PIT results for other methods:
> >
> > We appreciate the reviewer’s interest in uncertainty calibration. Calibration diagnostics such as PIT histograms and quantile calibration curves require access to the full conditional density, so they cannot be applied to regression-based models like XGBoost or DNN, which produce only point predictions. This limitation underscores an important advantage of conditional density estimators like CMLR and FlexCode.
> >
> > Motivated by the reviewer’s feedback, we have now:
> > - Expanded the calibration analysis to include FlexCode  (**Appendix A.9**)
> > - Added PIT histograms, quantile calibration curves, and ECE values (Appendix A.9; **Fig. S8**) of FlexCode
> > - Added a dedicated Calibration subsection in the main text (**Section 5.4**, p. 9)
> >
> > Across datasets, we found that CMLR exhibits near-uniform PIT distributions, low ECE, and well-behaved PIT–error relationships, whereas FlexCode exhibits systematic miscalibration (see Fig. S8). These results indicate that CMLR provides more reliable and interpretable uncertainty estimates, which further strengthen its merits. We thank the reviewer for this insightful comment.

---

> > > ### Author Response · Authors · 2025-11-20
> > >
> > > ## Q5.  Linking PIT values to decoding error:
> > >
> > > Inspired by the reviewer’s suggestion for a more dataset-specific evaluation, we added PIT–error plots in **Appendix A.9** (**Fig. S8C**). These analyses show that:
> > > - For CMLR, decoding errors are smallest near PIT $\approx 0.5$ and increase smoothly toward 0 and 1. This pattern indicates that CMLR indeed assigns higher uncertainty precisely in regions where decoding is intrinsically more difficult, such as the endpoints of the hippocampal track, and this behavior is consistent across datasets.
> > > - For FlexCode, errors peak near PIT $\approx 0.5$ and exhibit irregular structure, suggesting that its diffuse predictive densities do not reliably reflect variations in predictive difficulty.
> > >
> > > These results directly link calibration performance to meaningful structure in the datasets and highlight an additional practical advantage of CMLR as a conditional density estimation method.
> > >
> > > ## Q6. Clarifying “MLR-like decoding methods”:
> > >
> > > As suggested, we revised the **Introduction** to clarify this terminology. In particular, we now explain that “MLR-like decoding methods’’ refer to classification-style probabilistic decoders that output normalized probability distributions over discrete classes, distinct from standard regression methods, which output point predictions and cannot naturally handle multimodal or circular continuous variables.
> > >
> > > # Concluding Remarks
> > >
> > > We thank the reviewer again for the insightful suggestions, which have significantly improved the manuscript. The revised submission now includes:
> > > - Full hyperparameter tuning for all baselines via cross-validated Bayesian optimization
> > > - Comprehensive scalability and runtime analyses
> > > - Expanded uncertainty calibration and PIT–error analyses
> > > - Clearer exposition of CMLR’s complexity, design choices, and conceptual motivation
> > > - Improved Introduction and methodological clarity
> > >
> > > We hope that these substantial revisions address all of the reviewer’s concerns, and we believe that the updates further strengthen the clarity, rigor, and practical value of the proposed CMLR framework.
> > >
> > > ## References
> > >
> > > Gardner, J., Kusner, M., Xu, Z., Weinberger, K., & Cunningham, J. (2014). Bayesian optimization with inequality constraints. ICML, 937–945.
> > >
> > > Glaser, J. et al. (2020). Machine learning for neural decoding. eNeuro, 7(4).

---

> > > > ### Comment · Reviewer_4VSC · 2025-11-23
> > > >
> > > > I appreciate the authors' comprehensive updates in the new manuscript, and believe these have strengthened the work considerably. Accordingly, I have increased my score.
> > > >
> > > > Minor: Is the R2 for naive Bayes in Fig. 4C actually negative?

---

> > > > > ### Author Response · Authors · 2025-11-23
> > > > >
> > > > > Thank you very much for the updated assessment and for increasing your score; we sincerely appreciate it. We are glad to hear that the revisions strengthened the paper.
> > > > >
> > > > > Regarding your minor question: yes, the $R^2$ value for Naive Bayes in Fig. 4C is indeed negative. This reflects the fact that Naive Bayes performs substantially worse than a simple baseline predictor (the mean velocity), resulting in a negative coefficient of determination. As shown visually in Fig. 4B, Naive Bayes has the largest decoding error, and this is also evident in the Euclidean error values in the box plot (Fig. 4C). Quantitatively, DNN achieves the best performance (mean $\pm$ std: 4.2 $\pm$ 3.5), followed by XGBoost (4.3 $\pm$ 3.7), CMLR (4.4 $\pm$ 3.8), and FlexCode (4.7 $\pm$ 4.6), with Naive Bayes performing substantially worse (7.3 $\pm$ 7.0). This large error is directly reflected in its negative $R^2$. The poor performance is expected in this dataset: Naive Bayes assumes conditional independence across neurons, which is strongly violated in motor cortex, where population activity exhibits rich correlations and low-dimensional structure. Ignoring these correlations causes Naive Bayes to severely misestimate the likelihood, leading to large errors and, consequently, a negative $R^2$.
> > > > >
> > > > > Thank you again for your thoughtful engagement with our work and for helping us improve the manuscript. Your detailed comments directly strengthened the clarity, rigor, and positioning of the paper, and we are sincerely grateful for the time and care you invested in the review.

---

### Official Review · Reviewer_R8W8 · 2025-11-01

**Soundness:** 3
**Presentation:** 3
**Contribution:** 2
**Rating:** 4
**Confidence:** 2

**Summary:**

This paper introduces CMLR, a generalization of classical multinomial logistic regression to handle continuous output variables. Instead of discrete class weights, CMLR defines smooth, output-dependent weight functions with GP priors, allowing it to model conditional probability densities over continuous variables such as time, orientation, or spatial position. The authors derive a SVI algorithm in the Fourier domain for scalable learning on large datasets. They show CMLR’s performance on diverse neural decoding tasks across multiple brain regions (mouse and monkey V1, hippocampus CA1, motor cortex), and show that it outperforms Naive Bayes, FlexCode, XGBoost, and DNN in both accuracy and calibration.

**Strengths:**

1. Although it relies on strong modeling assumptions than nonlinear methods, the approach is flexible enough to handle circular and multidimensional outputs.

2. The method is interpretable, learning weight functions that correspond to tuning curves.

3. This method offers an attractive alternative for researchers who prefer to avoid the complexity and hyperparameter tuning required by nonlinear models while still getting strong decoding performance.

**Weaknesses:**

The DNN results in Figs 2 and 3 show relatively poor performance with large variance. While I understand that DNNs typically have higher variance than the proposed method, the degree of underperformance here suggests that the architecture or hyperparameter tuning might not have been well optimized for these decoding tasks. This raises some concern about the fairness of the comparison. It would be helpful for the authors to acknowledge this limitation in the paper. If, on the other hand, the proposed method achieves strong performance without requiring extensive hyperparameter tuning, that could be an additional advantage of the proposed method.

Relatedly, I would encourage the authors to clarify how they see the contribution of this work in the context of increasingly expressive modern models and large-scale neuroscience datasets. Although the proposed method is more interpretable, it may lag behind nonlinear models (e.g., transformers) in decoding performance. From a practical standpoint, the utility of the method might be limited. I would appreciate a clarification of how the authors envision their approach complementing or coexisting with more complex models.

**Questions:**

Despite using SVI for scalability, the method may still be computationally expensive for large-scale or high-dimensional datasets. A theoretical or empirical runtime comparison with baselines would strengthen the paper.

---

> ### Author Response · Authors · 2025-11-20
>
> We thank the reviewer for the thoughtful evaluation and for highlighting the strengths of CMLR, including its interpretability, flexibility, and its practical appeal for situations where researchers prefer to avoid the complexity and tuning burden of nonlinear models. We address the concerns and questions point-by-point below.
>
> # DNN performance and fairness of comparison
>
> We agree that the DNN performance in the original Figs. 2 and 3 showed high variance and that the hyperparameter procedure was not fully documented. In the revised manuscript, we have addressed this by:
> - performing **full 5-fold cross-validation** for all methods and datasets,
> - optimizing XGBoost and DNN hyperparameters via **Bayesian optimization** [Gardner et al., 2014] within each fold, using a 64–16–20 train/validation/test split, following best practices from Glaser et al. (2020),
> - **searching over key hyperparameters**:
> 1. XGBoost: tree depth, number of trees, learning rate
> 2. DNN: number of hidden units, dropout rate, number of epochs.
>
> The procedure is now clearly described in the revised manuscript (pp. 5–6), and the updated **Figs. 2, 3, 4, and S6** reflect the tuned results. CMLR continues to outperform XGBoost and DNN on Mouse V1 (Fig. 2), Monkey V1 (Fig. S6), and Mouse CA1 (Fig. 3), though with somewhat reduced performance gaps. These datasets all fall in regimes where *sample sizes are moderate relative to dimensionality* and where the output variable exhibits a *bounded structure*. In such settings, CMLR benefits from its **GP-based functional priors and additive structure**, which provide strong regularization, promote smoothness in the decoded variable, and mitigate overfitting when data are limited. In contrast, XGBoost and DNNs require larger sample sizes to fully leverage their capacity and tend to be more sensitive to noise and trial-to-trial variability in these datasets.
>
> Furthermore, the comparatively larger performance gap observed for circular orientation decoding (Mouse V1 and Monkey V1) can be attributed to several additional factors:
> - **CMLR explicitly models the bounded and circular nature of orientation**, whereas XGBoost and DNN do not incorporate circular geometry.
> - **Conditional density estimation (CDE) provides full predictive distributions**, allowing CMLR to represent periodic and multimodal structure (and we observed underlying bimodality in these datasets). Point-estimate models cannot capture these features.
> - **XGBoost and DNN performance depends critically on incorporating activity from adjacent time bins** (as in Mouse CA1 and Monkey MC). In the V1 datasets, however, adjacent orientations are not continuous by the design of the experiment, limiting the benefit of temporal or contextual information for these models.
>
> We now explicitly discuss these points in the revised manuscript (**Introduction**, p. 2, lines 86–92).
>
> For Monkey MC (Fig. 4), XGBoost and DNN now slightly outperform CMLR. We feel this is perhaps unsurprising given the very large training set that favors high-capacity nonlinear models. (We thank the reviewer for the suggestion to carefully optimize the hyperparameters for these two methods, which has indeed changed the relative ordering of methods for this dataset!)  To further analyze dataset-size effects, we have added a new scalability study on the Mouse V1 dataset (**Appendix A.6**, Fig. S5), which shows that CMLR experiences only modest reductions in accuracy as the number of neurons $D$ and samples $N$ decrease while continuing to scale efficiently. In these lower-data regimes, the performance of CMLR improves substantially relative to XGBoost and DNN. These updated results provide a clearer and fairer comparison across methods and illustrate the settings in which CMLR offers the greatest advantage. We thank the reviewer for highlighting the importance of these adjustments and believe they significantly strengthen the robustness of our findings.

---

> > ### Author Response · Authors · 2025-11-20
> >
> > # CMLR design parameters
> >
> > We appreciate the reviewer highlighting that CMLR requires minimal hyperparameter tuning. Unlike data-driven methods, CMLR does not require extensive dataset-specific hyperparameter tuning. We have added a new section (**Appendix A.5**) that provides practical guidance for selecting all CMLR design parameters and clarifies that these settings influence numerical precision and computational efficiency rather than model capacity. Specifically, the appendix now explains:
> >
> > 1. **Riemann bins $T$:** how $T$ determines the resolution of the normalization integral, with principled guidelines for selecting $T$ based on output-range width (e.g., $T \approx 100$ for outputs in $[0,1]$).
> > 2. **Fourier components $M$:** how $M$ controls the spectral resolution of the GP prior, why only low-frequency components are needed for RBF kernels, and why values in the range $M \in[15,50] $ provide stable performance while keeping computation low.
> > 3. **Mini-batch size $N’$:** how it affects optimization speed rather than model capacity, with practical ranges $(1000–2000)$ that balance runtime and memory.
> > 4. **Learning rate $\alpha$:** how $\alpha$ governs optimizer stability, with guidance that moderately small values $\alpha \in [10^{-3}, 10^{-1}]$ provide reliable convergence.
> > 5. **Decoding grid resolution $J$:** how $J$ determines the evaluation grid for predictive densities, with final results reported at the finest resolution tested ($J=20,000$).
> >
> > Appendix A.5 also includes **Table 2**, which reports the specific parameter settings used for each real-data experiment. Finally, the **Discussion** now notes that nonlinear baselines typically require extensive hyperparameter tuning, whereas CMLR’s design parameters require minimal dataset-specific adjustment, which is a practical advantage of the framework.
> >
> > # Contribution relative to expressive modern models
> >
> > We appreciate the reviewer’s question about CMLR’s role in the broader landscape of modern neural decoding models. In the revised **Discussion** (p. 9, lines 476–485), we clarify that:
> > - **Highly expressive nonlinear models**, such as deep neural networks and tree-based ensembles (and potentially transformers), can achieve superior predictive accuracy when very large training sets are available. This is indeed reflected in the Monkey MC dataset, where both DNN and XGBoost exceed CMLR’s performance due to the substantial sample size.
> > - **CMLR fills a distinct niche**: it provides fully probabilistic and well-calibrated conditional densities, is data-efficient, requires only a small number of design parameters (with no dataset-specific hyperparameter tuning), exhibits stable performance across datasets, and is grounded in a well-understood exponential-family framework.
> > - **Interpretability is a core advantage of CMLR**. Unlike black-box models, CMLR produces an explicit decoding weight function $w_d(y)$ for each neuron, which acts as a continuous tuning curve over the output domain and can be directly visualized and compared across neurons in the same manner as classical systems-neuroscience tuning curves.
> > - **As a conditional density estimation method**, CMLR provides richer information than point-estimate regressors, including multimodality, skewness, circular structure, and uncertainty quantification.
> > - **We therefore view CMLR as a practical, interpretable baseline or diagnostic model** that naturally complements more complex nonlinear approaches, especially in settings where uncertainty, interpretability, or data efficiency are important.
> >
> > # Responses to specific questions
> >
> > ## Q1. Computational cost and scalability:
> >
> > We thank the reviewer for requesting a runtime analysis. In the revised manuscript, we now provide a detailed assessment of computational efficiency, including:
> > - an explicit discussion of how **CMLR’s complexity scales with key parameters** (**Section 3.3**, p. 4, lines 204–211),
> > - a full **runtime comparison** across all models (**Section 5.5**, Table 1), and
> > - an additional **scalability analysis** in **Appendix A.6** (Fig. S5) showing how runtime and accuracy scale with the number of neurons $D$ and samples  $N$.
> >
> > Across datasets, CMLR trains within minutes to a few hours, with runtimes comparable to FlexCode and substantially faster than Naive Bayes. Although XGBoost and DNNs are typically faster, they do not produce full conditional densities or calibrated uncertainties. The scaling study further shows that CMLR remains efficient and robust across a range of data sizes.

---

> > > ### Author Response · Authors · 2025-11-20
> > >
> > > # Concluding remarks
> > >
> > > We sincerely appreciate the reviewer’s thoughtful comments and suggestions. In response, the revised manuscript provides:
> > > full hyperparameter tuning for all baselines,
> > > - clearer exposition of CMLR’s design and advantages,
> > > - an expanded discussion situating CMLR within the landscape of modern models, and
> > > - detailed runtime and scalability comparisons.
> > >
> > > Together, we believe that these additions substantially strengthen the empirical rigor of the work and further clarify the practical value and complementary role of CMLR in modern neural decoding. We hope that the revisions adequately address the reviewer’s concerns. If the reviewer feels that these updates satisfactorily address the issues raised in the original review, we would greatly appreciate it if the reviewer would be willing to update their score.
> > >
> > > ## References
> > >
> > > Gardner, J., Kusner, M., Xu, Z., Weinberger, K., & Cunningham, J. (2014). Bayesian optimization with inequality constraints. ICML, 937–945.
> > >
> > > Glaser, J. et al. (2020). Machine learning for neural decoding. eNeuro, 7(4).

---

> > > > ### Comment · Reviewer_R8W8 · 2025-11-21
> > > >
> > > > Thank you for running the hyperparameter tuning and runtime comparison experiments to address my concerns. I personally prefer more expressive and scalable models for large datasets, which I believe will be the future trend in neuroscience. However, the proposed method does have strengths in small-data settings and provides a probabilistic framework that is useful for uncertainty quantification. Accordingly, I have raised my score to 6. I hope my comments have helped improve the paper.

---

> > > > > ### Author Response · Authors · 2025-11-21
> > > > >
> > > > > Thank you very much for your thoughtful response and for raising your score. We sincerely appreciate it. We are grateful for the constructive feedback throughout the review process; your comments directly strengthened the clarity, rigor, and positioning of the paper. We fully agree that increasingly expressive and scalable models will play a crucial role in future neural decoding work. We view CMLR as a complementary tool that offers calibrated uncertainty estimates and strong performance in small-data and structured-output settings. Thank you again for your thoughtful engagement and for helping us improve the manuscript.

---

### Official Review · Reviewer_LDws · 2025-11-07

**Soundness:** 3
**Presentation:** 3
**Contribution:** 4
**Rating:** 6
**Confidence:** 3

**Summary:**

This paper develops a novel approach, Continuous Multinomial Logistic Regression (CMLR), that is an extension of MLR to predict continuous-valued outputs, and showcase this approach for neural decoding (predicting behaviors/stimuli from neural activity). The model enables interpretable understanding of smooth 'tuning curves' of neurons within the decoding model via GP priors. They develop an approach to fit the model with stochastic variational inference. The authors demonstrate excellent neural decoding performance, in additional to interpretability of the model across multiple datasets using their method.

**Strengths:**

This is an original, creative, and novel technical development for neural decoding (and prediction problems more broadly). The method has a good blend of accuracy and interpretability (via the learned tuning curves within the model). The paper is clearly written and the authors are very thorough. It is a very strong paper overall.

**Weaknesses:**

My one fundamental concern is with the comparisons in results, specifically in terms of hyperparameter selection. I might have missed it, but it's not clear to me how/if hyperparameter tuning was done both for the author's model and for comparison models. It seems odd how poor many of the results are from what should be close to state-of-the-art approaches (XGBoost and DNNs), which makes me suspect poor hyperparameters (causing either overfitting or underfitting). Proper hyperparameter tuning on a held-out validation set (within the training set) should be done, if it wasn't already.

**Questions:**

1. Fig 4C - Euclidean error is hard to interpret - Coefficient of determination (the standard r2_score in sklearn) would be easier to interpret, and more standard for velocity decoding

2. Discussion: "First, unlike models such as XGBoost or DNNs, CMLR does not incorporate priors over outputs" - I don't understand this statement in terms of XGBoost and DNNs having priors on outputs.

3. Line 263 - closing bracket ] missing

4. I would put the methods in section 5.3 into an actual methods section. They feel very out of place when reading results

5. How long does your approach take to run compared to others?

---

> ### Author Response · Authors · 2025-11-20
>
> We sincerely thank the reviewer for the positive and thoughtful evaluation, and for recognizing the novelty and interpretability of the proposed CMLR framework. Below, we address the reviewer’s main concern regarding hyperparameter tuning and fairness of comparisons, and respond to each question in turn.
>
> # Hyperparameter tuning and fairness of comparisons
>
> We appreciate the reviewer raising this important point, and we acknowledge that the original manuscript did not clearly describe this procedure. In the revised version, we have addressed this by:
> - performing **full 5-fold cross-validation** for all methods and datasets,
> - optimizing XGBoost and DNN hyperparameters via **Bayesian optimization** [Gardner et al., 2014] within each fold, using a 64–16–20 train/validation/test split, following best practices from Glaser et al. (2020),
> - **searching over key hyperparameters**:
> 1. XGBoost: tree depth, number of trees, learning rate
> 2. DNN: number of hidden units, dropout rate, number of epochs.
>
> The procedure is now clearly described in the revised manuscript (pp. 5–6), and the updated **Figs. 2, 3, 4, and S6** reflect the tuned results. CMLR continues to outperform XGBoost and DNN on Mouse V1 (Fig. 2), Monkey V1 (Fig. S6), and Mouse CA1 (Fig. 3), though with somewhat reduced performance gaps. These datasets all fall in regimes where *sample sizes are moderate relative to dimensionality* and where the output variable exhibits a *bounded structure*. In such settings, CMLR benefits from its **GP-based functional priors and additive structure**, which provide strong regularization, promote smoothness in the decoded variable, and mitigate overfitting when data are limited. In contrast, XGBoost and DNNs require larger sample sizes to fully leverage their capacity and tend to be more sensitive to noise and trial-to-trial variability in these datasets.
>
> Furthermore, the comparatively larger performance gap observed for circular orientation decoding (Mouse V1 and Monkey V1) can be attributed to several additional factors:
> - **CMLR explicitly models the bounded and circular nature of orientation**, whereas XGBoost and DNN do not incorporate circular geometry.
> - **Conditional density estimation (CDE) provides full predictive distributions**, allowing CMLR to represent periodic and multimodal structure (and we observed underlying bimodality in these datasets). Point-estimate models cannot capture these features.
> - **XGBoost and DNN performance depends critically on incorporating activity from adjacent time bins** (as in Mouse CA1 and Monkey MC). In the V1 datasets, however, adjacent orientations are not continuous by the design of the experiment, limiting the benefit of temporal or contextual information for these models.
>
> We now explicitly discuss these points in the revised manuscript (Introduction, p. 2, lines 86–92).
>
> For Monkey MC (Fig. 4), XGBoost and DNN now slightly outperform CMLR. We feel this is perhaps unsurprising given the *very large training set* that favors *high-capacity nonlinear models*. (We thank the reviewer for the suggestion to carefully optimize the hyperparameters for these two methods, which has indeed changed the relative ordering of methods for this dataset!)  To further analyze dataset-size effects, we have added a new scalability study on the Mouse V1 dataset (**Appendix A.6**, Fig. S5), which shows that CMLR experiences only modest reductions in accuracy as the number of neurons $D$ and samples $N$ decrease while continuing to scale efficiently. In these lower-data regimes, the performance of CMLR improves substantially relative to XGBoost and DNN. These updated results provide a clearer and fairer comparison across methods and illustrate the settings in which CMLR offers the greatest advantage. We thank the reviewer for highlighting the importance of these adjustments and believe they significantly strengthen the robustness of our findings.

---

> > ### Author Response · Authors · 2025-11-20
> >
> > # CMLR design parameters
> >
> > Thank you for noting that the original description was unclear. Unlike data-driven methods, CMLR does not require extensive dataset-specific hyperparameter tuning. We have added a new section (**Appendix A.5**) that provides practical guidance for selecting all CMLR design parameters and clarifies that these settings influence numerical precision and computational efficiency rather than model capacity. Specifically, the appendix now explains:
> >
> > 1. **Riemann bins $T$:** how $T$ determines the resolution of the normalization integral, with principled guidelines for selecting $T$ based on output-range width (e.g., $T \approx 100$ for outputs in $[0,1]$).
> > 2. **Fourier components $M$:** how $M$ controls the spectral resolution of the GP prior, why only low-frequency components are needed for RBF kernels, and why values in the range $M \in[15,50] $ provide stable performance while keeping computation low.
> > 3. **Mini-batch size $N’$:** how it affects optimization speed rather than model capacity, with practical ranges $(1000–2000)$ that balance runtime and memory.
> > 4. **Learning rate $\alpha$:** how $\alpha$ governs optimizer stability, with guidance that moderately small values $\alpha \in [10^{-3}, 10^{-1}]$ provide reliable convergence.
> > 5. **Decoding grid resolution $J$:** how $J$ determines the evaluation grid for predictive densities, with final results reported at the finest resolution tested ($J=20,000$).
> >
> > Appendix A.5 also includes **Table 2**, which reports the specific parameter settings used for each real-data experiment. Finally, the **Discussion** now notes that nonlinear baselines typically require extensive hyperparameter tuning, whereas CMLR’s design parameters require minimal dataset-specific adjustment, which is a practical advantage of the framework.
> >
> > # Responses to specific questions
> >
> > ## Q1: Use of  $R^2$ for velocity decoding:
> >
> > We thank the reviewer for this helpful suggestion and fully agree. We now report $R^2$  in Fig. 4C for improved interpretability and consistency with prior decoding work.
> >
> > ## Q2: “Priors over outputs’’ in DNN/XGBoost:
> >
> > We thank the reviewer for pointing out the unclear wording. What we intended to convey is that, following Glaser et al. (2020), DNNs and XGBoost often incorporate neural activity from multiple adjacent time bins when predicting the current output, thereby implicitly enforcing temporal continuity in the decoded variable. Our current implementation of CMLR does not yet include such temporal or output-space structure, which could be useful in applications such as navigation or motor decoding. We have clarified this point in the revised **Discussion** (p. 10, lines 515–517).
> >
> > ## Q3. Closing bracket:
> >
> > We confirm that the output range is $[0,2\pi)$. The right boundary is open to avoid duplication of 0 and $2 \pi$.
> >
> > ## Q4. Misplaced methods in Section 5.3:
> >
> > We thank the reviewer for this helpful suggestion. We have moved the relevant methodological details to the Methods section in the revised manuscript (p. 4-5).
> >
> > ## Q5. Run time comparisons:
> >
> > Thanks for this suggestion!  We have now added runtime comparisons for all methods in newly added **Section 5.5** (**Table 1**, p. 9). CMLR trains within minutes to a few hours depending on dataset size, with runtimes comparable to FlexCode, and substantially faster than Naive Bayes. XGBoost and DNNs are generally somewhat faster, but do not produce full conditional densities or calibrated uncertainty. Additional runtime scaling analyses (Appendix A.6, Fig. S5) show that CMLR remains efficient under varied data configurations.
> >
> > # Concluding remarks:
> >
> > We thank the reviewer once again for the constructive feedback and insightful suggestions. The revised manuscript now incorporates:
> > - full hyperparameter tuning for all baselines,
> > - clarified CMLR design choices,
> > - reorganized methodological material for improved clarity, and
> > - expanded runtime and scalability analyses.
> >
> > We hope these revisions fully address the reviewer’s concerns, and we believe they substantially strengthen the fairness and clarity of the comparisons while further highlighting the robustness and practical value of CMLR.
> >
> > ### References
> >
> > Gardner, J., Kusner, M., Xu, Z., Weinberger, K., & Cunningham, J. (2014). Bayesian optimization with inequality constraints. ICML, 937–945.
> >
> > Glaser, J. et al. (2020). Machine learning for neural decoding. eNeuro, 7(4).

---

> > > ### Comment · Reviewer_LDws · 2025-11-27
> > >
> > > Thank you for the rigorous additional comparisons - I'm increasing my score.
> > >
> > > I do have one followup clarification question:
> > > In your response to Q2, you state "DNNs and XGBoost often incorporate neural activity from multiple adjacent time bins when predicting the current output".  In all of the experiments in your paper, did the DNNs and XGBoost use a single time bin of data (so that the input features are identical as CMLR)?

---

> > > > ### Author Response · Authors · 2025-11-27
> > > >
> > > > Thank you very much for the updated assessment and for increasing your score; we sincerely appreciate it.
> > > >
> > > > Thank you also for the follow-up question. In all of the experiments, the DNN and XGBoost models did **not** use a single time bin as input. Instead, they incorporated multiple adjacent time bins from both the past and the future. We made this choice for two reasons. First, these models benefit considerably from temporal context, and in preliminary experiments their performance with a single time bin was much lower than what is typically reported for high-capacity nonlinear decoders. Second, we wanted the comparison to reflect the intrinsic strengths of each method rather than placing DNN or XGBoost in a configuration that does not match their usual and most effective usage. By contrast, CMLR was evaluated with a single time bin because the current implementation does not naturally support multi-time-bin inputs and is designed for instantaneous decoding.
> > > >
> > > > This difference is also reflected in the results. XGBoost and DNN performance depends critically on incorporating activity from adjacent time bins, as in the Mouse CA1 and Monkey MC datasets. In the V1 datasets, however, adjacent orientations are not continuous by the design of the experiment, which limits the benefit of temporal or contextual information for these models.

---

> > > > > ### Comment · Reviewer_LDws · 2025-11-27
> > > > >
> > > > > Thanks for the clarification. I think the choice you made allows for a fairer comparison, as you said. I would make sure these general points are clear in the final paper (if they're not mentioned already).

---

> > > > > > ### Author Response · Authors · 2025-11-27
> > > > > >
> > > > > > Thank you for highlighting this point; we fully agree. In the revised manuscript, we have explicitly clarified that XGBoost and DNN were trained using multiple adjacent time bins as input (p. 6, lines 285–295).
> > > > > >
> > > > > > We are grateful for your continued engagement and for the insightful comments provided throughout the review process. Your feedback has substantially strengthened the manuscript, and we sincerely appreciate the time and care you devoted to the evaluation.

---

### Meta-Review · Area_Chair_Gm8Q · 2026-01-08

**Summary:**

This submission proposes Continuous Multinomial Logistic Regression (CMLR), a principled extension of multinomial logistic regression to continuous outputs via output-dependent weight functions with Gaussian process priors, fit using stochastic variational inference in the Fourier domain. A key strength, repeatedly emphasized by reviewers, is the method’s combination of strong decoding performance with interpretability, where learned weight functions can be visualized as smooth neuron-specific “tuning-like” curves, and where the model outputs full conditional densities enabling uncertainty quantification and calibration analyses.

Across reviews, the primary recurring concern was the rigor of baseline comparisons, especially whether DNN and XGBoost were appropriately tuned and whether results were robust to the train/test split. Reviewers also requested clearer reporting of runtime/complexity, more explicit guidance for CMLR design parameters, and improved presentation choices (e.g., reporting R2 for velocity decoding, moving misplaced methods content). In addition, there were questions how CMLR’s interpretability differs from feature importance in XGBoost and how the method is situated relative to more expressive modern models on large datasets which may outperform CMLR in terms of raw decoding accuracy.

The authors’ rebuttal and revision were responsive to the critiques. They implemented a 5-fold cross-validation and performed Bayesian hyperparameter optimization within each fold for DNN/XGBoost following established practice. They added a dedicated treatment of computational complexity and runtime comparisons, plus scaling studies that highlight CMLR’s data-efficiency advantages in lower-data settings. The revision strengthened the paper’s conceptual positioning of CMLRas a complementary tool rather than as a replacement for the most expressive decoders in the high-data regime.

Overall, the authors make a compelling case for CMLR as an interpretable, probabilistic decoder for neural data, which enables uncertainty quantification and calibration analyses.  The version post-rebuttal appears substantially strengthened in baseline fairness and validation/explanation of the interpretability of the method. The contributions of the work to neural data analysis and consensus amongst the reviewers point in favor of acceptance.

**Reviewer Concerns:**

No major concerns were left unaddressed after the rebuttal.

**Reviewer Scores:**

Reviewer LDws and Reviewer R8W8 noted that they intended to increase their scores from 6 to 8, and from 4 to 6 respectively. Reviewer 4VSC raised their score from 6 to 8 after noting that the revisions satisfactorily addressed points 1 through 4. Reviewer Z69w gave an initial score of 6 and indicated that points 1 through 5 were addressed, and thus likely maintained the same score.

---

### Decision · Program_Chairs · 2026-01-26

Accept (Poster)